# TENSOR-GALORE: MEMORY-EFFICIENT TRAINING VIA GRADIENT TENSOR DECOMPOSITION

## ABSTRACT

We present **Tensor-GaLore**, a novel method for efficient training of neural networks with higher-order tensor weights. Many models, particularly those used in scientific computing, employ tensor-parameterized layers to capture complex, multidimensional relationships. When scaling these methods to high-resolution problems makes memory usage grow intractably, and matrix based optimization methods lead to suboptimal performance and compression. We propose to work directly in the high-order space of the complex tensor parameter space using a tensor factorization of the gradients during optimization. We showcase its effectiveness on Fourier Neural Operators (FNOs), a class of models crucial for solving partial differential equations (PDE) and prove the theory of it. Across various PDE tasks like the Navier Stokes and Darcy Flow equations, Tensor-GaLore achieves substantial memory savings, reducing optimizer memory usage by up to 75%. These substantial memory savings across AI for science demonstrate Tensor-GaLore's potential.

## 1 INTRODUCTION

The advent of foundation models has revolutionized AI, demonstrating unprecedented performance across diverse domains such as natural language processing, computer vision, and scientific computing Brown et al. (2020); Kirillov et al. (2023). However, as these models grow in scale and complexity, they present significant computational challenges. With parameters often numbering in the billions, these models demand enormous memory resources for storage and optimization, making their training and deployment prohibitively expensive for many researchers and organizations. Recent work (See Section 6) has focused on parameter-efficient fine-tuning and pre-training methods to address these issues.

Gradients in deep neural networks often exhibit low-rank structures during training, implying that the most important gradient information can be stored at a fraction of the memory cost. GaLore (Gradient Low-Rank Projection) Zhao et al. (2024) leveraged this insight to reduce memory usage in large language model training by projecting large gradients onto low-rank subspaces and optimizing on the low-rank gradients. Specifically, GaLore operates on weight matrices $W \in \mathbb{R}^{m \times n}$ and their corresponding gradient matrices $G \in \mathbb{R}^{m \times n}$. For a given rank $r$, GaLore computes the Singular Value Decomposition (SVD) of the gradient matrix, forms projection matrices using the first $r$ singular vectors, then projects the gradient onto this low-rank subspace to perform optimization. After computing the optimizer update, the gradients are projected back to their full rank for use in the model. This approach allows GaLore to maintain a low memory footprint by storing and updating only the low-rank representations of gradients.

However, GaLore's approach is limited to matrix operations and relies on Singular Value Decomposition (SVD), which may not be optimal for all neural network layers or data structures. In particular, GaLore faces significant challenges when applied to "tensor" operations, which are prevalent in many modern deep learning architectures, especially those used in scientific computing and computer vision. *Tensors* are multidimensional arrays that offer a natural framework for representing and manipulating complex, high-dimensional data structures, and the limitations of matrix-based approaches like GaLore when applied to tensor operations is the fact that many models involve inherently tensor-structured gradients, where preserving the multidimensional relationships is crucial for capturing complex physical phenomena. Simply flattening or "matricizing" these tensors

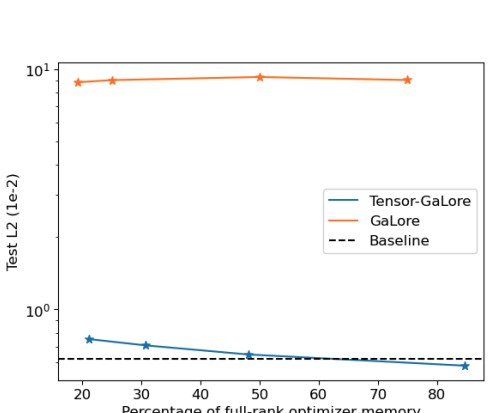
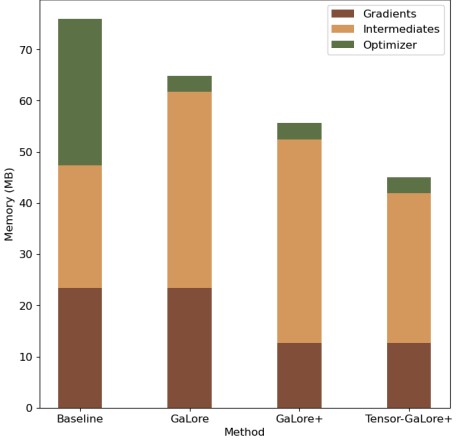

Figure 1: Left: Test $L_2$ results on Navier-Stokes (128 resolution) vs. optimizer memory usage. Right: CUDA memory usage for FNO models on Navier-Stokes. GaLore+ includes per-layer optimization and activation checkpointing. Tensor-GaLore significantly reduces memory usage.

into matrices can lead to a loss of important dimension-specific information and may not allow for maximum compression and memory savings. In certain architectures, different dimensions might correspond to spatial, temporal, or channel information, each requiring distinct treatment that is best preserved in the original tensor form.

The field of scientific modeling has seen a significant paradigm shift towards applying AI to it. Neural operators (NOs) Li et al. (2020) is one of the most promising new architectures in this domain. The neural operator is a framework for modeling multi-scale processes on continuous domains. Having the discretization invariance property, the operator learns a continuum mapping, allowing NOs to model systems that traditional neural networks cannot accurately capture. The ability to handle multi-scale processes on continuous domains represents a key advantage of NOs over conventional neural network approaches in scientific modeling. FNOs are a class of neural operator architecture designed to learn mappings between function spaces to solve parametric PDEs, a cornerstone of modern scientific computing.

Unlike traditional neural networks, FNOs involve 4th-order or 5th-order tensor operations. In an FNO, the spectral convolution layer contracts a weight tensor $\mathcal{R} \in \mathbb{C}^{N_1 \times N_2 \times N_3 \times N_4}$ with functions in the Fourier domain: $(\mathcal{K}v_l)(x) = \mathcal{F}^{-1}(R\mathcal{F}v_l)(x)$, where $\mathcal{F}$ and $\mathcal{F}^{-1}$ are the Fourier transform and its inverse, $R$ is a learnable transformation parameterized by the weight tensor introduced above.

While these tensor operations are powerful for capturing complex, high-dimensional relationships in scientific data, they pose unique challenges related to memory consumption during training. The primary issue lies not in the activation memory induced by forward and backward passes but in the memory overhead required for optimization. This overhead is due to the need to store the Fourier coefficients and perform operations in the frequency domain Lingsch et al. (2024). This memory bottleneck is further exacerbated by modern optimizers, which often store multiple tensors for each weight tensor to track gradients, momentum, and other quantities, as in the case of Adam. Consequently, the optimizer state comprises a significant portion of the memory overhead in training large-scale NOs. As illustrated in Figure 5, the memory consumption for activations (shown in dark green) remains relatively constant and low across different numbers of frequency modes in FNOs. However, the memory usage for individual components, including gradients and optimizer states (shown in yellow), grows significantly as the number of modes increases. Increasing frequency modes is crucial for capturing finer details in complex systems like turbulent fluids. Still, it comes at the cost of higher memory usage, presenting a key challenge in scientific machine learning.

These challenges motivate the need for a tensor-specific approach to gradient projection and optimization. Hence we introduce **Tensor-GaLore, a novel method for efficiently training NOs through low-rank gradient projections**. To the best of our knowledge, Tensor-GaLore is the first work to explore low-rank subspace learning for gradients of higher-order tensors that seeks low-rank representation while offering a significant advancement in memory-efficient optimization and topologically preserving the structure. Tensor-GaLore utilizes Tucker decomposition to project gra-

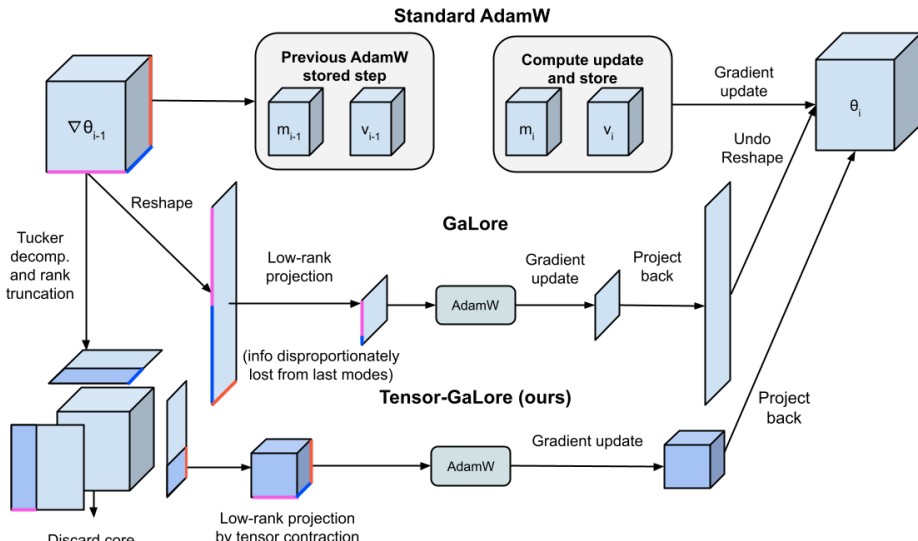

Figure 2: Comparison of our proposed Tensor-GaLore algorithm with standard AdamW and Ga-Lore. GaLore applies matrix-based low-rank projection after reshaping tensors. Our Tensor-GaLore method leverages tensor decomposition to perform low-rank projection directly on tensor gradients, preserving multidimensional structure.

dient tensors onto low-rank subspaces, preserving the multidimensional structure crucial for NOs as shown in Figure 2. Additionally, we also prove the theory of Tensor-GaLore including convergence and low-rank gradient behaviour of these tensor weights.

We demonstrate the effectiveness of Tensor-GaLore on a diverse set of PDE tasks, with our largest case study focusing on the Navier-Stokes equations at 1024x1024 resolution. For this computationally intensive problem, our experiments show significant reductions in memory usage (up to 75% for optimizer states). Figure 4 illustrates these substantial memory savings across different ranks. In addition, we validate Tensor-GaLore's performance on other important PDEs such as Darcy flow, Burgers' equation, and electromagnetic wave propagation.

Tensor-GaLore opens new possibilities for developing and deploying advanced AI systems across various scientific and engineering disciplines by enabling more efficient training of large-scale tensor-based models. Our approach democratizes access to large-scale Neural Operator training, allowing researchers with limited computational resources to work on cutting-edge problems in scientific computing and AI-driven scientific discovery.

## 2 BACKGROUND: GALORE AND NEURAL OPERATORS

### 2.1 NEURAL OPERATOR

A neural operator $\mathcal{G}_\theta : \mathcal{A} \times \theta \to \mathcal{U}$ combines linear integral operators $\mathcal{K}$ with pointwise non-linear activations $\sigma$ to approximate non-linear operators, mapping initial conditions $a \in \mathcal{A}$ to solutions $u \in \mathcal{U}$. Its operation is defined as $\mathcal{G}_\theta := \mathcal{Q} \circ (W_L + \mathcal{K}_L) \circ \cdots \circ \sigma(W_1 + \mathcal{K}_1) \circ \mathcal{P}$, where $\mathcal{P}$ and $\mathcal{Q}$ are pointwise neural networks for encoding and decoding, $W_l$ are linear operators, $\mathcal{K}_l$ are integral kernel operators, and $\sigma$ are activation functions.

The Fourier Neural Operator (FNO) proposes a specific convolution operator for $\mathcal{K}$, defined as $(\mathcal{K}v_l)(x) = \mathcal{F}^{-1}(R \cdot T_K \mathcal{F} v_l)(x)$, where $\mathcal{F}$ and $\mathcal{F}^{-1}$ are the Fourier transform and its inverse, $R$ is a learnable transformation, and $T_K$ truncates to the lowest $K$ Fourier modes. This formulation allows FNO to be discretization-invariant, producing high-quality solutions for query points not in the training grid and enabling transfer between different grid resolutions and discretizations.

## 2.2 Challenges of applying GaLore to neural operators

In order to apply standard GaLore to tensor weights, the weights must first be reshaped into a matrix to compute the SVD for projection into a low-rank space. GaLore takes one rank parameter, $r$, and projects high-rank gradients onto the first $r$ basis vectors of the corresponding SVD rotation matrix. When the weight matrix corresponds to an operator that maps between vectors, a single rank cutoff can be applied while preserving most information.

However, in the tensor case, weights correspond to higher-order maps between function spaces. Depending on the chosen strategy for reshaping tensor weights into a matrix, applying a single-dimension rank cutoff to the matrix may discard key information - for instance, for a tensor $W \in \mathbb{C}^{A \times B \times m \times m}$, where $A$ is the number of input channels, $B$ is the number of output channels, and $m$ is the number of truncated Fourier basis modes along each dimension, reshaping $W$ into $W' \in \mathbb{C}^{ABm \times m}$ and cutting off the first dimension at rank $r$ may remove all information about Fourier modes along the first dimension, making function learning impossible. We call this method *GaLore* and provide several comparisons to demonstrate its flaws.

One flaw is the **Loss of mode-specific information**: by collapsing multiple tensor dimensions into one matrix dimension, we lose the ability to preserve different amounts of information along each tensor mode. The other is that we have an **imbalanced projection**: Projecting only on one side of the reshaped matrix (e.g., only $U$ or only $V$ from the SVD) can severely limit the operator's capacity. However, projecting on both sides often leads to training instability and failure to converge. This method also encounters **rank selection issues**: Choosing a single rank cutoff for the reshaped matrix makes it difficult to balance information preservation across all the original tensor dimensions. A rank that preserves enough information for one dimension may be too restrictive for another.

## 3 Tensor-GaLore

### 3.1 Tensor Decomposition

Tensors are multidimensional arrays that generalize the concepts of vectors (first-order tensors) and matrices (second-order tensors) to higher orders. An $N$th-order tensor $\mathcal{X} \in \mathbb{C}^{I_1 \times I_2 \times \cdots \times I_N}$ is an $N$-way array where each mode $n$ has dimension $I_n$. Like matrices, in tensors, we can decompose the tensors into low-rank factors using the Tucker decomposition, also known as the higher-order SVD (HOSVD), which decomposes a tensor into a core tensor multiplied by a matrix along each mode:

$$\mathcal{X} \approx \mathcal{G} \times_1 U^{(1)} \times_2 U^{(2)} \cdots \times_N U^{(N)} = [\![\mathcal{G}; U^{(1)}, U^{(2)}, \ldots, U^{(N)}]\!] \tag{1}$$

where $\mathcal{G} \in \mathbb{C}^{R_1 \times R_2 \times \cdots \times R_N}$ is the core tensor, $U^{(n)} \in \mathbb{C}^{I_n \times R_n}$ are factor matrices, and $\times_n$ denotes the $n$-mode product. Two critical aspects of the Tucker decomposition make it particularly suitable for our Tensor-GaLore method:

1. **Equivalence to SVD in 2D**: In the special case of 2D tensors (matrices), the Tucker decomposition reduces to the familiar SVD. The core tensor $\mathcal{G}$ becomes equivalent to the diagonal matrix $\Sigma$ in SVD, while the factor matrices correspond to the orthogonal matrices $U$ and $V$ Kolda & Bader (2009). This property ensures that our method seamlessly extends the principles of matrix-based techniques to higher-order tensors.

2. **Orthogonality of factor matrices**: The factor matrices $U^{(n)}$ in Tucker decomposition are orthogonal, mirroring the properties of $U$ and $V$ in SVD. This orthogonality is crucial for the efficiency and stability of the GaLore method. Specifically:

(a) *Projection efficiency*: The orthogonality allows us to project tensors onto the subspace spanned by these matrices through simple matrix multiplication, without the need for costly inverse computations.

(b) *Easy inversion*: When we need to reverse the projection, we can simply use the transpose of these orthogonal matrices instead of computing their inverses. This property is expressed mathematically as $(U^{(n)})^T U^{(n)} = I$, where $I$ is the identity matrix.

(c) *Numerical stability*: Orthogonal matrices have a condition number of 1, ensuring that the projection and its inverse are numerically stable operations, even for high-dimensional tensors.

We use TensorLy's Kossaifi et al. (2019) implementation of Tucker decomposition, which is based on Higher-Order Orthogonal Iteration (HOI). For an input tensor $X$, HOI computes approximate values for the Tucker factor matrices $\{U^{(i)}\}_i$ by approximating the SVD of the unfolding of $X$ along each mode. HOI updates these factors iteratively to minimize the Frobenius norm between $X$ and the resulting learned decomposition. These learned factors can be initialized with nonzero values, meaning that once full HOI is computed once, the decomposition can be "warm-restarted" to reduce the number of iterations required for convergence.

In addition to these steps, like in GaLore, we incorporate per-layer weight updates Lv et al. (2024) and activation checkpointing Chen et al. (2016) to reduce memory usage further. Per-layer weight updates allow the optimizer to update weights immediately after computing gradients for each layer rather than storing gradients for all layers before updating. This method reduces the peak memory requirement during training. Activation checkpointing involves selectively recomputing certain activations during the backward pass instead of storing them, trading some additional computation for reduced memory usage. Combined with low-rank gradient projection, these techniques enable Tensor-GaLore to achieve significant memory savings while maintaining training efficiency and performance. We denote this method as Tensor-GaLore/GaLore +.

**Extension:** To extend GaLore to methods with learned tensor weights, we replace the matrix-based SVD with tensor decomposition methods. This extension, called Tensor-GaLore, allows us to handle multi-dimensional data and complex network architectures more efficiently.

---

**Algorithm 1** Adam with Tensor-GaLore

---

**Require:** A layer weight tensor $\mathcal{W} \in \mathbb{C}^{N_1 \times N_2 \times N_3 \times N_4}$. Step size $\eta$, scale factor $\alpha$, decay rates $\beta_1, \beta_2$, rank $r$, subspace change frequency $T$.
1: Initialize first-order moment $\mathcal{M}_0 \in \mathbb{C}^{N_1 \times N_2 \times N_3 \times N_4} \leftarrow 0$
2: Initialize second-order moment $\mathcal{V}_0 \in \mathbb{C}^{N_1 \times N_2 \times N_3 \times N_4} \leftarrow 0$
3: Initialize step $t \leftarrow 0$
4: **repeat**
5:     $\mathcal{G}_t \in \mathbb{C}^{N_1 \times N_2 \times N_3 \times N_4} \leftarrow -\nabla_{\mathcal{W}} \phi_t(\mathcal{W}_t)$
6:     **if** $t \bmod T = 0$ **then**
7:         $\mathcal{C}, \{U^{(n)}\}_{n=1}^4 \leftarrow \text{Tucker}(\mathcal{G}_t, \text{rank} = r)$            ▷ Initialize projector.
8:     **else**
9:         $\mathcal{C}, \{U^{(n)}\}_{n=1}^4 \leftarrow \mathcal{C}_{t-1}, \{U_{t-1}^{(n)}\}_{n=1}^4$      ▷ Reuse the previous projector.
10:    **end if**
11:    $\mathcal{R}_t \leftarrow \mathcal{G}_t \times_1 U^{(1)\top} \times_2 U^{(2)\top} \times_3 U^{(3)\top} \times_4 U^{(4)\top}$   ▷ Project gradient into compact space.
12:    **UPDATE**$(\mathcal{R}_t)$ by Adam:
13:       $\mathcal{M}_t \leftarrow \beta_1 \cdot \mathcal{M}_{t-1} + (1 - \beta_1) \cdot \mathcal{R}_t$
14:       $\mathcal{V}_t \leftarrow \beta_2 \cdot \mathcal{V}_{t-1} + (1 - \beta_2) \cdot |\mathcal{R}_t \bar{\mathcal{R}}_t|$      ▷ We use the complex conjugate update.
15:       $\mathcal{M}_t \leftarrow \mathcal{M}_t / (1 - \beta_1^t)$
16:       $\mathcal{V}_t \leftarrow \mathcal{V}_t / (1 - \beta_2^t)$
17:       $\mathcal{N}_t \leftarrow \mathcal{M}_t / (\sqrt{\mathcal{V}_t} + \epsilon)$
18:    $\tilde{\mathcal{G}}_t \leftarrow \alpha \cdot \mathcal{N}_t \times_1 U^{(1)} \times_2 U^{(2)} \times_3 U^{(3)} \times_4 U^{(4)}$     ▷ Project back to original space.
19:    $\mathcal{W}_t \leftarrow \mathcal{W}_{t-1} + \eta \cdot \tilde{\mathcal{G}}_t$
20:    $t \leftarrow t + 1$
21: **until** convergence criteria met.
22: **return** $\mathcal{W}_t$

---

For a gradient tensor $\mathcal{G} \in \mathbb{C}^{I_1 \times I_2 \times \cdots \times I_N}$, the Tucker-based Tensor-GaLore performs the following steps:

    1. Compute the Tucker decomposition of the gradient tensor:

$$\mathcal{G} \approx \mathcal{C} \times_1 U^{(1)} \times_2 U^{(2)} \cdots \times_N U^{(N)} = [\![\mathcal{C}; U^{(1)}, U^{(2)}, \ldots, U^{(N)}]\!] \tag{2}$$

    where $\mathcal{C} \in \mathbb{C}^{R_1 \times R_2 \times \cdots \times R_N}$ is the core tensor and $U^{(n)} \in \mathbb{C}^{I_n \times R_n}$ are factor matrices.

2. Project the gradient tensor onto the low-rank subspace and update the optimizer states and model parameters using the projected gradient $\mathcal{G}_{\text{proj}}$.

$$\mathcal{G}_{\text{proj}} = [\![G_{\text{core}}U^{(1)^T}, U^{(2)^T}, \ldots, U^{(N)^T}]\!] \tag{3}$$

3. Project the gradient back when updating.

$$\mathcal{G}_{\text{core}} = [\![G_{\text{proj}}U^{(1)}, U^{(2)}, \ldots, U^{(N)}]\!] \tag{4}$$

## 3.2 THEORETICAL RESULTS OF TENSOR-GALORE

We extend the theoretical foundations of GaLore to tensor-structured weights, proving both convergence guarantees and low-rank emergence during training. Our analysis shows that gradients of FNO models naturally develop low-rank structure in each tensor mode during training, while Tensor-GaLore achieves convergence through mode-wise projections. All the proofs and background details are in Appendix sections H, I and J.

**Theorem 1 (Tensor-GaLore Convergence)** *For a gradient tensor $\mathcal{G}_t \in \mathbb{R}^{I_1 \times I_2 \times \cdots \times I_d}$, let $\{P_k \in \mathbb{R}^{I_k \times r_k}\}_{k=1}^d$ be fixed orthonormal projection matrices for each mode $k$ with ranks $\{r_k\}_{k=1}^d$. Suppose for each mode $k$:*

- *$\mathcal{A}_i$, $\mathcal{B}_i$, $\mathcal{C}_i$ have $L_A^{(k)}$, $L_B^{(k)}$, $L_C^{(k)}$ mode-k continuity, $\|\mathcal{W}_t\|_{(k)} \leq D_k$ (mode-k spectral norm bound), $\hat{\mathcal{B}}_{it}^{(k)} := P_k^\top \mathcal{B}_i^{(k)}(\mathcal{W}_t)P_k$, $\hat{\mathcal{C}}_{it}^{(k)} := P_k^\top \mathcal{C}_i^{(k)}(\mathcal{W}_t)P_k$ , $\kappa_t^{(k)} := \frac{1}{N}\sum_i \lambda_{\min}(\hat{\mathcal{B}}_{it}^{(k)})\lambda_{\min}(\hat{\mathcal{C}}_{it}^{(k)})$*

*Then Tensor-GaLore with $\rho_t \equiv 1$ satisfies for each mode $k$:*

$$\|(\mathcal{R}_t)_{(k)}\|_F \leq \left[1 - \eta(\kappa_{t-1}^{(k)} - L_A^{(k)} - L_B^{(k)} L_C^{(k)} D_k^2)\right] \|(\mathcal{R}_{t-1})_{(k)}\|_F$$

*As a result, if $\min_{t,k} \kappa_t^{(k)} > L_A^{(k)} + L_B^{(k)} L_C^{(k)} D_k^2$ for all modes $k$, then $\mathcal{R}_t \to 0$ and Tensor-GaLore converges with the fixed projections $\{P_k\}_{k=1}^d$. Proof 10.*

**Remark 1 (Mode-k Continuity)** *The mode-k continuity assumption on $\mathcal{A}_i$, $\mathcal{B}_i$, $\mathcal{C}_i$ is mild and holds generically for neural network parameters.*

## 3.3 IMPLICIT REGULARIZATION

Tucker decomposition is defined with a separate rank along each mode of the decomposed tensor, preserving all key information explicitly. Additionally, the resulting decomposition's factors can be initialized to non-random values in Tucker decomposition. As learning progresses, results from a previous decomposition can be used to 'warm-restart' the process, leading to convergence in fewer iterations.

The low-rank tensor approximation acts as an implicit regularizer, helping to prevent overfitting and promoting smoother optimization trajectories. Hence, we observe much better convergence and generalization in our experiments. In particular, we consistently observed that a rank of around 25% - 50% of the total rank provided optimal performance across various tasks. This observation suggests that Tensor-GaLore acts as an implicit regularizer, preventing overfitting by constraining the model to learn more robust, low-rank representations of the underlying physics. These results align with findings from Razin et al. (2022), demonstrating that tensor factorization naturally tends towards low-rank solutions. In our experiments, we saw dramatic improvements in convergence even with a fixed number of epochs, sometimes achieving over 50% improvement in test loss. This result implies that the regularization effect might be even more significant in higher-order tensors due to the increased structure and redundancy in these higher-dimensional spaces.

## 4 EXPERIMENTAL SETUP

We conduct a comprehensive evaluation of GaLore and Tensor-GaLore on a diverse set of benchmark datasets for NOs. We select datasets representing a range of PDEs with varying complexity and dimensionality. These include:

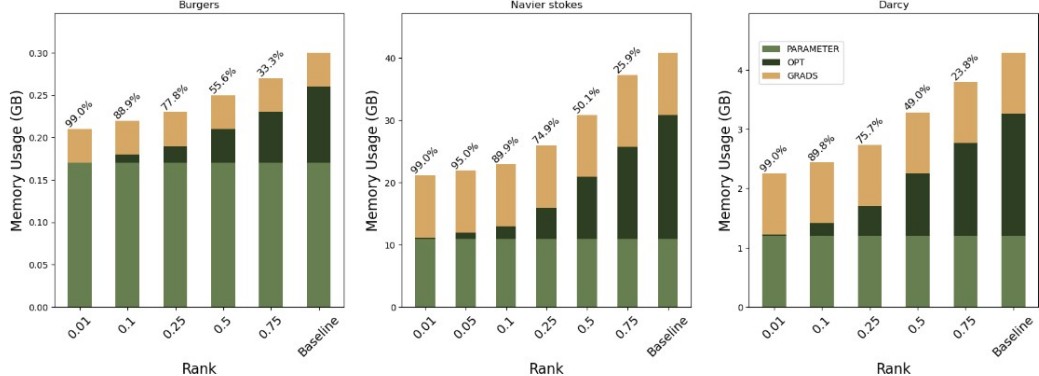

Figure 3: Memory usage of FNO and GINO on various datasets on an NVIDIA A100. On top of the bars, we showcase the reduction in optimizer memory in % using Tensor-GaLore.

## 4.1 DATASETS

**Burgers Equation:**   We consider the one-dimensional Burgers equation on the torus:

$$\partial_t u + u u_x = \nu u_{xx}, \quad x \in \mathbb{T}, t \in (0, T] \tag{5}$$

with initial condition $u_0 \in L^2(\mathbb{T}; \mathbb{C})$ and viscosity $\nu > 0$. We set $T = 1$ and $\nu = 0.01$. Input functions are sampled from a Gaussian random field, and solutions are obtained using a pseudo-spectral method. We use 1000 samples for training and 200 for testing, with 128 resolution.

**Navier-Stokes:**   We use the two-dimensional Navier-Stokes equation in vorticity form:

$$\partial_t \omega + \nabla^\perp \phi \cdot \omega = \frac{1}{Re} \Delta \omega + f, \quad x \in \mathbb{T}^2, t \in (0, T]$$
$$-\Delta \phi = \omega, \quad \int_{\mathbb{T}^2} \phi = 0, \quad x \in \mathbb{T}^2, t \in (0, T] \tag{6}$$

with Reynolds number $Re = 1000$ and final time $T = 5$. The domain is discretized on a 1024 × 1024 grid. We generate 10000 training samples and 2000 test samples using a pseudo-spectral method. We also showcase the effectiveness of our approach at a subsampled resolution of 128 × 128. Our memory profiling is also done at the full 1024 × 1024 resolution.

**Darcy Flow:**   The Darcy flow problem is defined by the elliptic PDE:

$$-\nabla \cdot (a(x) \nabla u(x)) = f(x), \quad x \in (0, 1)^2 \tag{7}$$

with boundary conditions $u(x) = 0$ for $x \in \partial(0, 1)^2$. The input $a$ is sampled from a Gaussian random field, and $f$ is fixed. We use 4000 training samples and 1000 test samples, with the domain discretized on a 421 × 421 grid.

**Electromagnetic Wave Propagation:**   Lastly, we present a dataset that represents complex-valued data inherently. We consider the propagation of optical pulses in a nonlinear waveguide with second-order nonlinearity ($\kappa^2$). The problem is governed by the nonlinear Schrödinger equation (NLSE) with additional terms for second-harmonic generation:

$$\frac{\partial A}{\partial z} = -i \frac{\beta_2}{2} \frac{\partial^2 A}{\partial t^2} + i\gamma |A|^2 A + i\kappa A^* e^{i\Delta k z} \tag{8}$$

where $A$ is the complex electric field envelope, i is the imaginary unit, $z$ is the propagation distance, $t$ is time, $\beta_2$ is the group velocity dispersion, $\gamma$ is the nonlinear parameter, $\kappa$ is the coupling coefficient

for second-harmonic generation, and $\Delta k$ is the phase mismatch. Our dataset consists of 800 training samples and 200 testing samples. The input consists of several parameters: the poling region length ranging from 2mm to 15mm, the poling period mismatch varying from -50nm to +50nm, and the pump pulse energy spanning from a few fJ to thousands of fJ. Additionally, the input includes the complex electric field envelope of the input pulse. The output of the system is the complex electric field envelope of the resulting output pulse.

## 4.2 MODEL ARCHITECTURE AND TRAINING

We implement Tensor-GaLore with the FNO architecture. Models are trained using an AdamW optimizer. Other training details, such as learning rate, batch size, epochs, losses, are detailed in Appendix 8 for each dataset and model configuration. [1].

For Tensor-GaLore, we investigate the impact of varying the rank of the decompositions. We explore ranks ranging from 20% to 100% of the total rank, allowing us to assess the trade-off between model compression and performance. We explore comparable matrix ranks for GaLore to provide a direct comparison with our method. Detailed results for these ablations are provided in Appendix D. Additionally, we explore various ways of reshaping the tensor to a matrix for tensor inputs before applying GaLore. Specifically, we examine each possible "matricization" dimension, where we flatten multiple tensor dimensions into a single matrix dimension. This allows us to compare the effectiveness of different tensor-to-matrix projections. Details are in Appendix D.

**Evaluation Metrics** We evaluate our models using the $L_2$ and $H_1$ loss to provide a comprehensive assessment of performance. In PDE's the $H_1$ loss, accounts for both the function values and their gradients, providing a more rigorous assessment of the solution's smoothness and accuracy. The gain percentage is calculated based on the improvement in $L_2$ test loss compared to the baseline.

Table 1: Evaluating Tensor-GaLore across various tasks.

| Model | Rank Ratio | Memory (GB) | Train (Loss ($\times 10^{-2}$)) | Test $H_1$ (Loss ($\times 10^{-2}$)) | Test $L_2$ (Loss ($\times 10^{-2}$)) | Gain (%) |
|---|---|---|---|---|---|---|
| **Darcy** | | | | | | |
| Baseline | 1.0 | 8.88 | 0.7151 | 1.6230 | 0.2050 | / |
| GaLore (d=2) | 0.25 | 7.34 | 0.4200 | 1.3210 | 0.1680 | 19 |
| Tensor-GaLore | 0.25 | 7.32 | **0.2930** | **0.8680** | **0.1050** | **48.8** |
| **Navier-Stokes** | | | | | | |
| Baseline | 1.0 | 77 | **1.0630** | **1.9010** | **0.6152** | / |
| GaLore (d=1) | 0.5 | 68 | 4.3340 | 5.5830 | 1.9952 | -223 |
| Tensor-GaLore | 0.5 | 55 | 1.2340 | 2.0850 | 0.6480 | -5.4 |
| **ElectroMagnetic** | | | | | | |
| Baseline | 1.0 | 4.83 | 2.973 | 0.1902 | 0.2000 | / |
| GaLore (d=2) | 0.25 | 4.83 | 2.392 | 0.1802 | 0.1900 | 5 |
| Tensor-GaLore | 0.25 | 4.63 | **2.132** | **0.1681** | **0.1782** | **11** |
| **Burgers** | | | | | | |
| Baseline | 1.0 | 3.94 | 0.0064 | 0.0050 | 0.0026 | / |
| GaLore (d=2) | 0.5 | 3.88 | 0.0052 | 0.0100 | 0.0062 | -250 |
| Tensor-GaLore | 0.5 | 3.87 | **0.0026** | **0.0041** | **0.0025** | **+5** |

## 5 RESULTS

Our experiments demonstrate the effectiveness of Tensor-GaLore across various datasets, showing significant improvements in both performance and memory efficiency as shown in Table 1. For the Burgers equation, our method consistently outperformed the baseline FNO, with performance improving as rank increased. On the Darcy flow problem, Tensor-GaLore achieved up to a 50% gain in test loss at rank 0.25, while reducing optimizer memory

---

[1]Code is available at: https://anonymous.4open.science/r/tensorgalore

Table 2: Model performance on Darcy-flow.

| Model | Test Loss (1e-2) at Rank Ratio | | | | | | Gain (%) |
|---|---|---|---|---|---|---|---|
| | 0.01 | 0.1 | 0.25 | 0.5 | 0.75 | 1.0 | |
| FNO Baseline | - | - | - | - | - | 0.205 | / |
| FNO - Tensor-GaLore | **0.147** | **0.108** | **0.105** | **0.107** | **0.140** | **0.173** | **49** |
| FNO - GaLore (d=1) | 0.256 | 0.232 | 0.212 | 0.245 | 0.201 | 0.190 | 8 |
| FNO - GaLore (d=2) | 0.203 | 0.192 | 0.168 | 0.178 | 0.170 | 0.180 | 19 |
| FNO - GaLore (d=3) | 0.234 | 0.212 | 0.201 | 0.193 | 0.196 | 0.182 | 11 |

by 76%. The Navier-Stokes experiments showcased Tensor-GaLore's ability to handle complex problems, maintaining comparable performance at lower ranks while dramatically reducing memory usage. Electromagnetic wave propagation simulations saw up to 11% gains.

Across all tested datasets, Tensor-GaLore also demonstrated superior performance to GaLore at comparable ranks, suggesting that preserving higher-order structures within the weight gradients can substantially improve model performance. The results show that Tensor-GaLore can significantly reduce the memory footprint of the optimizer states while improving model performance in many cases. On Darcy flow (as shown in Table 5), we observed up to an 48% improvement in test loss with a rank of 0.25, while reducing the optimizer state memory from 2.09GB to 0.5GB. On Navier-Stokes, we achieve even more significant memory savings while achieving comparable performance to the baseline.

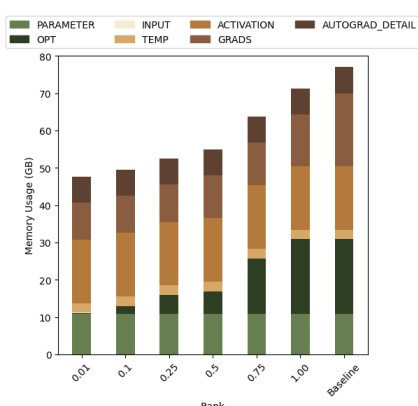

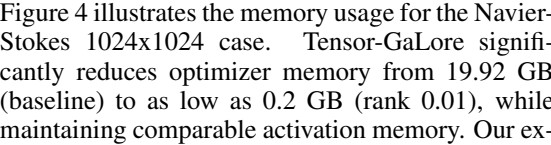

Figure 4 illustrates the memory usage for the Navier-Stokes 1024x1024 case. Tensor-GaLore significantly reduces optimizer memory from 19.92 GB (baseline) to as low as 0.2 GB (rank 0.01), while maintaining comparable activation memory. Our ex-

Figure 4: Memory usage of NS 1024 using an FNO on a A100. Comparison between Tensor-GaLore and baseline.

periments reveal a trend in performance gains across problem complexities. For simpler problems like Darcy flow, Tensor-GaLore achieves substantial improvements, but as problem complexity increases, such as with Navier-Stokes at 128x128 resolution, the performance gains become more modest but still significant. This pattern suggests that Tensor-GaLore's effectiveness scales with problem difficulty. We have a detailed parameter and memory complexity analysis in Appendix G.

## 6 RELATED WORK

Our work, Tensor-GaLore, introduces a novel approach to efficiently training neural operators by decomposing gradients. While significant work has been done in related areas, the specific approach of gradient decomposition in tensors has not been explored. **Tensor Methods in Deep Learning:** Tensor decomposition has been widely used to compress and improve deep networks, particularly in vision tasks Novikov et al. (2015); Lebedev et al. (2015); Kim et al. (2016). These methods typically focus on decomposing the weight tensors of the network to reduce parameters and computational complexity. However, they do not address the decomposition of gradients during training.

**Neural Operators:** Recent advancements in learning-based approaches for solving PDEs have led to the development of neural operators Li et al. (2020); Kovachki et al. (2021). In particular, FNOs have shown remarkable success in various scientific computing tasks Li et al. (2021). While these methods have made significant strides in learning solution operators for PDEs, they have not explored gradient decomposition to improve memory efficiency.

**Efficient Training Techniques:** Various approaches have been proposed to reduce the memory footprint of large-scale models. In the classical case, several techniques have demonstrated success when model weights are stored as matrices. LoRA Hu et al. (2022) adds a fine-tuning weight matrix created via a low-rank decomposition to an original pre-trained, frozen weight matrix. In the higher-order case, FLoRA Si et al. (2024) extends the idea of low-rank adaptation to higher-dimensional parameter spaces using a Tucker tensor decomposition, which has the demonstrated benefit of applying a low-rank decomposition to each dimension of a higher-order space. In the context of neural operators, which include higher-order tensorized weights, previous works have demonstrated the possibility of model compression via tensor factorization and low-rank weight approximations. Kossaifi et al. (2024) introduced the Multi-Grid Tensorized Fourier Neural Operator (MG-TFNO), which combines tensor decomposition with a multi-grid domain decomposition approach. In order to balance low-rank memory optimization with model performance at higher ranks, the Incremental Fourier Neural Operator (iFNO) George et al. (2024) incrementally scales both the size and rank of FNO weights during training in order to boost performance.

**Mixed Precision Training** Mixed precision training Tu et al. (2024) utilizes lower precision formats (e.g., FP16) for certain operations in NO, reducing memory usage and potentially accelerating training on compatible hardware.

Tensor-GaLore introduces a novel approach that can complement and enhance many existing techniques, potentially leading to even greater memory benefits. It can be combined with mixed precision training, integrated with methods like FLoRA or MG-TFNO to provide an additional layer of optimization for gradient tensors, and incorporated into frameworks like iFNO.

## 7 APPLICATIONS

Tensor-GaLore has potential applications across various domains where tensor-based models are prevalent. Large language models (LLMs) could enable the training of tensor-based architectures that capture higher-order relationships in language data, offering improved memory efficiency and implicit regularization while preserving the natural tensor structure. Convolutional Neural Networks (CNNs) also heavily utilize higher-order tensor weights in vision. CNN convolution layers include 4-dimensional tensor weights. As discussed previously, these weight gradients and optimizer states have high memory requirements, making memory consumption a significant bottleneck in training deep CNNs Yaqub et al. (2020). Future applications of Tensor-GaLore could scale these methods and improve their performance in constrained environments.

## 8 CONCLUSION

The results of our experiments with Tensor-GaLore reveal several key insights into its performance and potential applications. First, the consistent improvement in convergence across various datasets is noteworthy. By projecting gradients onto a low-rank subspace, Tensor-GaLore appears to create a more stable optimization landscape, potentially smoothing out local minima and facilitating faster convergence to better solutions. These results are particularly evident in the Darcy flow and Navier-Stokes experiments, where we observed improved test loss even at lower ranks. Additionally, the ability to warm-start each decomposition using factors from the previous iteration likely contributes to maintaining stable convergence despite frequent subspace changes. However, Tensor-GaLore has limitations. The overhead of performing tensor decomposition, while amortized, may still be significant for some applications, and the optimal rank selection remains a challenge that requires further investigation. Future work should focus on exploring the application of Tensor-GaLore to an even broader range of scientific computing tasks.

Lastly, Tensor-GaLore represents a significant advancement in memory-efficient training for large-scale tensor-based models, particularly in AI for Science. Tensor-GaLore opens up new avenues for building and scaling foundational models in scientific computing by enabling the training of more complex neural operators with dramatically reduced memory footprints. Our results demonstrate that this approach not only preserves performance but often enhances it, suggesting that the implicit regularization induced by low-rank projections may be particularly beneficial for capturing the underlying physics of complex systems. This could lead to more accurate and computationally efficient models for climate prediction, fluid dynamics, and other critical scientific applications.

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

# APPENDIX

## A    FNO MEMORY USAGE

Figure 5 illustrates the memory usage patterns in Fourier Neural Operators (FNOs) as the number of modes increases. This analysis provides crucial insights into the scalability challenges faced when training large FNO models.

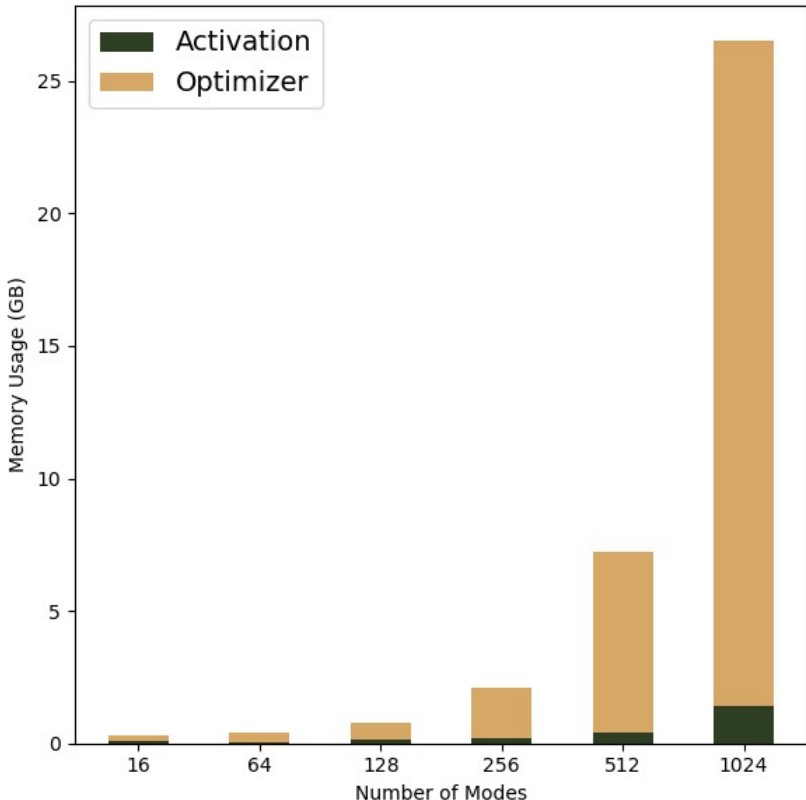

Figure 5: Memory usage in FNO as a function of the number of modes

As evident from the figure, the memory consumption is divided into two main categories: activation memory and optimizer memory. The activation memory, represented by the dark green bars, remains relatively constant and low across different numbers of modes. This stability in activation memory is a positive attribute of FNOs, indicating that the forward and backward passes do not significantly increase memory requirements as the model complexity grows.

However, the optimizer memory, shown in yellow, exhibits a dramatic increase as the number of modes grows. This exponential growth in optimizer memory becomes particularly pronounced for models with more than 128 modes. For instance, when the number of modes reaches 1024, the optimizer memory dominates the total memory usage, far exceeding the memory required for activations.

This trend highlights a critical bottleneck in scaling FNO models to higher resolutions or more complex problems. The optimizer's memory footprint, which includes storage for gradients, momentum, and adaptive learning rate parameters, becomes the primary limiting factor. This observation motivates the need for memory-efficient optimization techniques like Tensor-GaLore, which specifically target the reduction of optimizer memory usage while maintaining model performance.

## B  PROFILING METHODOLOGY

To analyze the performance and memory usage of our Tensor-GaLore method, we implemented a comprehensive profiling setup using PyTorch's built-in profiler. This allowed us to gain detailed insights into the computational and memory requirements of our algorithm compared to baseline methods.

**Detailed Memory Breakdown.** We implemented a detailed memory tracking system to distinguish between various types of memory usage, including Model parameters, Optimizer states, Input data, Activations, Gradients, Autograd details, Temporary buffers. To provide a comprehensive understanding of memory utilization in our experiments, we developed a classification system to distinguish between different types of memory usage. This granular approach allows us to precisely identify where memory savings occur when using Tensor-GaLore compared to baseline methods:

- **Model Parameters.** Model Parameters are udentified by tracking tensors that are registered as model parameters (instances of 'nn.Parameter'). It is typically constant throughout training unless using techniques like weight decay.
- **Optimizer States.** Optimizer States are tracked by instrumenting the optimizer to log memory allocations for momentum buffers, adaptive learning rate parameters, etc. For Adam optimizer, this includes first and second moment estimates.
- **Input Data.** Input is monitored by tracking memory allocations that occur during data loading and preprocessing steps.
- **Activations.** Activations are identified as temporary tensors created during the forward pass of the model. it is tracked using hooks on module forward methods to capture intermediate outputs.
- **Activations.** Activations are identified as temporary tensors created during the forward pass of the model. it is tracked using hooks on module forward methods to capture intermediate outputs.
- **Gradients.** Gradients ared recognized as tensors with 'requires_grad=True' that are outputs of operations on model parameters or inputs.
- **Autograd Details.** It is captured by profiling PyTorch's autograd engine internals, including memory used for storing computational graphs and intermediate results needed for backpropagation.
- **Temporary Buffers.** Temporary Buffers are short-lived tensors that are created and destroyed within a single operation or a small set of operations. For tensor-galore, it is often used in complex computations like FFTs or tensor decompositions within galore.

To implement this detailed profiling, we used a combination of PyTorch's memory-profiler, custom context managers, and function decorators. Key aspects of our implementation include:

- Wrapping key operations with context managers to track memory allocation and deallocation
- Using PyTorch hooks to monitor intermediate activations and gradients
- Instrumenting the optimizer to log memory usage for each parameter update
- Implementing custom memory tracking for Tensor-GaLore specific operations

The results of this analysis formed the basis for our discussions on memory efficiency in Sections 5 and 6 of the main paper, and provided the data for Figure 4, which illustrates the memory usage breakdown for different numbers of frequency modes in FNOs.

## C  GALORE

## D  ADDITIONAL RESULTS

We evaluate three approaches to matricizing a tensor gradient with shape $C_{in} \times C_{out} \times M_x \times M_y$. The first, which we call "rollout=1", combines the last 3 dimensions into one matrix dimension, resulting in a matrix of shape $C_{in} \times (C_{out} * M_x * M_y)$. The second, "rollout=2", combines the first two dimensions into the first matrix dimension and the last two dimensions into the second matrix dimension, resulting in a matrix of shape $(C_{in} * C_{out}) \times (M_x * M_y)$. The last, "rollout=3",

---

**Algorithm 2** GaLore

---

**Require:** A layer weight tensor $\mathcal{W} \in \mathbb{C}^{N_1 \times N_2 \times N_3 \times N_4}$. Step size $\eta$, scale factor $\alpha$, decay rates $\beta_1, \beta_2$, rank $r$, subspace change frequency $T$, chosen dimension $d$.

1: Initialize first-order moment $\mathcal{M}_0 \in \mathbb{C}^{N_1 \times N_2 \times N_3 \times N_4} \leftarrow 0$
2: Initialize second-order moment $\mathcal{V}_0 \in \mathbb{R}^{N_1 \times N_2 \times N_3 \times N_4} \leftarrow 0$
3: Initialize step $t \leftarrow 0$
4: **repeat**
5:     $\mathcal{G}_t \in \mathbb{C}^{N_1 \times N_2 \times N_3 \times N_4} \leftarrow -\nabla_{\mathcal{W}} \phi_t(\mathcal{W}_t)$
6:     $G_t^{(d)} \leftarrow \text{Reshape}(\mathcal{G}_t, (N_d, \prod_{i \neq d} N_i))$               ▷ Reshape tensor to matrix
7:     **if** $t \bmod T = 0$ **then**
8:         $U, \Sigma, V^\top \leftarrow \text{SVD}(G_t^{(d)})$                    ▷ Compute SVD
9:         $P \leftarrow V[:, :r]^\top$             ▷ Select $r$ right singular vectors
10:     **end if**
11:     $R_t \leftarrow G_t^{(d)} P^\top$             ▷ Project gradient into compact space
12:     **UPDATE**($R_t$) by Adam:
13:         $M_t \leftarrow \beta_1 \cdot M_{t-1} + (1 - \beta_1) \cdot R_t$
14:         $V_t \leftarrow \beta_2 \cdot V_{t-1} + (1 - \beta_2) \cdot |R_t|^2$
15:         $M_t \leftarrow M_t/(1 - \beta_1^t)$
16:         $V_t \leftarrow V_t/(1 - \beta_2^t)$
17:         $N_t \leftarrow M_t/(\sqrt{V_t} + \epsilon)$
18:     $\tilde{G}_t^{(d)} \leftarrow \alpha \cdot N_t P$            ▷ Project back to original space
19:     $\tilde{\mathcal{G}}_t \leftarrow \text{Reshape}(\tilde{G}_t^{(d)}, (N_1, N_2, N_3, N_4))$         ▷ Reshape back to tensor
20:     $\mathcal{W}_t \leftarrow \mathcal{W}_{t-1} + \eta \cdot \tilde{\mathcal{G}}_t$
21:     $t \leftarrow t + 1$
22: **until** convergence criteria met
23: **return** $\mathcal{W}_t$

---

combines the last three dimensions into the second matrix dimension, resulting in a matrix of shape $C_{in} \times (C_{out} * M_x * M_y)$. We showcase results and comparisons for all three approaches in Table 6.

All of the subsequent results are with varying rank ratios on the Tensor-GaLore method for all datasets. We report both the training and testing loss/accuracy.

Table 3: Model performance on Burgers

| Model | Rank Ratio | Train Loss (1e-4) | Test Loss(1e-4) | Gain (%) |
|---|---|---|---|---|
| FNO Baseline | Full Rank | 0.205 | 0.262 | / |
| FNO - Tensor-GaLore | 0.1 | 0.115 | 0.321 | -19 |
| FNO - Tensor-GaLore | 0.25 | 0.095 | 0.271 | -4 |
| FNO - Tensor-GaLore | 0.5 | 0.086 | 0.253 | +5 |
| FNO - Tensor-GaLore | 0.75 | 0.083 | 0.246 | +8 |
| FNO - Tensor-GaLore | 1.00 | 0.083 | **0.242** | **+9** |

Table 4: Model performance on Darcy-flow

| Model | Rank Ratio | Train Loss (1e-2) | Test Loss(1e-2) | Gain (%) |
|---|---|---|---|---|
| FNO Baseline | Full Rank | 0.715 | 0.205 | / |
| FNO - Tensor-GaLore | 0.01 | 0.465 | 0.147 | +30 |
| FNO - Tensor-GaLore | 0.1 | 0.323 | 0.108 | +48 |
| FNO - Tensor-GaLore | 0.25 | **0.293** | **0.105** | **+49** |
| FNO - Tensor-GaLore | 0.5 | 0.275 | 0.107 | +49 |
| FNO - Tensor-GaLore | 0.75 | 0.379 | 0.140 | +40 |
| FNO - Tensor-GaLore | 1.00 | 0.715 | 0.173 | +16 |

Table 5: Model performance on EM.

| Model | Test Loss (1e-2) at Rank Ratio | | | | | | Gain (%) |
|---|---|---|---|---|---|---|---|
| | 0.01 | 0.1 | 0.25 | 0.5 | 0.75 | 1.0 | |
| FNO Baseline | - | - | - | - | - | 0.200 | / |
| FNO - Tensor-GaLore | 0.187 | 0.185 | **0.178** | 0.176 | 0.174 | 0.206 | **11** |
| FNO - GaLore (d=1) | 0.213 | 0.192 | 0.193 | 0.189 | 0.194 | 0.200 | 7 |
| FNO - GaLore (d=2) | 0.205 | 0.206 | 0.195 | 0.196 | 0.201 | 0.199 | 3 |

Table 6: Ablation: GaLore and Tensor-GaLore Rank Comparison

| Method | % orig. parameters | GaLore Test $L_2$ ($\times 10^{-2}$) | Tensor-GaLore Test $L_2$ ($\times 10^{-2}$) |
|---|---|---|---|
| GaLore (d=1) | 25 | 2.3410 | **1.2970** |
| | 50 | 1.9950 | **0.9982** |
| | 75 | 4.7530 | **0.9409** |
| GaLore (d=2) | 25 | - | **1.2970** |
| | 50 | 9.9800 | **0.9980** |
| | 75 | 0.1250 | **0.9409** |
| GaLore (d=3) | 25 | 9.0190 | **1.2970** |
| | 50 | 9.2390 | **0.9982** |
| | 75 | 9.0250 | **0.9409** |

Table 7: Model performance on EM

| Model | Rank Ratio | Train Loss | Test Loss | Gain (%) |
|---|---|---|---|---|
| Complex FNO Baseline | Full Rank | 2.973 | 0.200 | / |
| Complex FNO - Tensor-GaLore | 0.01 | 4.198 | 0.249 | -20 |
| Complex FNO - Tensor-GaLore | 0.1 | 2.936 | 0.217 | -8 |
| Complex FNO - Tensor-GaLore | 0.25 | **2.132** | **0.178** | +11 |
| Complex FNO - Tensor-GaLore | 0.5 | 2.430 | 0.184 | +8 |
| Complex FNO - Tensor-GaLore | 0.75 | 2.719 | 0.192 | +4 |
| Complex FNO - Tensor-GaLore | 1.00 | 2.397 | 0.185 | +8 |

# E ARCHITECTURE AND TRAINING DETAILS

**Sobolev Loss for PDE Training**   In training NOs for PDEs we employ both the $L^2$ and Sobolev $H^1$ losses to provide a comprehensive assessment of model performance. While the $L^2$ loss measures point-wise accuracy of predictions, the $H^1$ loss, defined as $\|u - \hat{u}\|_{H^1}^2 = \|u - \hat{u}\|_{L^2}^2 + \|\nabla u - \nabla \hat{u}\|_{L^2}^2$, accounts for both the function values and their gradients. This is particularly crucial for PDEs, as it ensures that the learned solutions not only match the target values but also preserve the smoothness and differential properties inherent in the physical systems being modeled.

**Sobolev Loss for Complex Wave Phenomena**   The Sobolev $H^1$ loss proves especially valuable when dealing with complex wave phenomena, as demonstrated in our experiments with the EM Dataset using Complex-FNOs. In this case, the $H^1$ loss not only measures the accuracy of the predicted complex electric field envelope but also ensures that its spatial derivatives are correctly captured. This is crucial for accurately representing the rapid oscillations and sharp peaks characteristic of EM waves. Our results show that Tensor-GaLore with a rank ratio of 0.25 achieved an 11% improvement in overall test loss compared to the baseline, with the $H^1$ loss decreasing from 0.1902 to 0.1681. This improvement is particularly significant given the challenging nature of the EM dataset, which involves predicting the complex electric field envelope resulting from nonlinear interactions in waveguides. The enhanced performance in $H^1$ loss indicates that our model not only matches the amplitude of the EM waves more accurately but also better captures the rapid spatial variations and peak formations. This is critical in applications such as optical pulse propagation, where precise modeling of field gradients and peak intensities is essential for predicting phenomena like second-harmonic generation and phase matching.

| Dataset | Model | Architecture Details | Optimizer & Scheduler |
|---|---|---|---|
| Burgers | FNO | <ul><li>4 layers, 90 modes</li><li>256 hidden channels, 256 projection channels</li><li>Skip Connections: 'linear'</li><li>Positional embedding: 'grid'</li></ul> | Adam with step LR $3e - 4$, weight decay $2e - 6$ 500 epochs, batch size 16. Trained with $H_1$ loss. |
| NS128 | FNO | <ul><li>4 layers, 64 x 64 modes</li><li>64 hidden channels, 256 projection channels</li><li>Skip: 'linear'</li><li>Use channel MLP: 1</li><li>Channel MLP expansion: 0.5, dropout: 0</li></ul> | Adam with step LR 3e-4, weight decay 1e-4, 500 epochs, batch size 8. Trained with $H_1$ loss. |
| NS1024 | FNO | <ul><li>4 layers, 100 modes</li><li>256 hidden channels, 256 projection channels</li><li>Skip: 'linear'</li></ul> | Adam with step LR |
| Darcy Flow | FNO | <ul><li>4 layers, 64 modes</li><li>128 hidden channels, 128 projection channels</li><li>Skip: 'linear'</li></ul> | Adam with step LR $1e - 3$, weight decay $1e - 4$, 250 epochs, batch size 2. Trained with $L_2$ loss. |
| EM Wave | Complex-FNO | <ul><li>8 layers, 128 modes</li><li>128 hidden channels, 128 projection channels</li><li>Skip: 'linear'</li><li>Complex data: True</li><li>Complex activation function: True</li></ul> | Complex Adam with step LR 1e-4, weight decay 2e-6, batch size 32, 1000 epochs. Trained with $H_1$ loss. |

Table 8: Detailed FNO Architecture Specifications for Different Datasets

## F   SLOWDOWN IN TRAINING

While Tensor-GaLore does introduce additional computational overhead from the tensor decomposition step, we have carefully analyzed the impact on training speed and efficiency. Our experiments have shown that the memory savings achieved by Tensor-GaLore often outweigh the slight increase in computational cost, resulting in an overall improvement in training time and resource utilization. Specifically, we have measured the training time for Tensor-GaLore compared to the baseline FNO model and the GaLore approach. Our results indicate that the slowdown in training time is modest, typically in the range of 5-20%, depending on the dataset and model configuration. This is a reasonable trade-off given the significant memory savings (up to 75% reduction in optimizer memory) that Tensor-GaLore provides.

| Model | Rank | Time/epoch(s) | Slowdown (%) |
|---|---|---|---|
| Baseline | 1.0 | 34.96 | – |
| GaLore | 0.20 | 34.47 | -1.40 |
| GaLore | 0.25 | 34.79 | -0.48 |
| GaLore | 0.50 | 36.27 | 3.75 |
| GaLore | 0.75 | 37.50 | 7.26 |
| Tensor-GaLore (40, 40, 40, 24) | 0.20 | 36.53 | 5.98 |
| Tensor-GaLore (48, 48, 48, 24) | 0.25 | 38.30 | 10.08 |
| Tensor-GaLore (56, 56, 56, 24) | 0.50 | 40.63 | 12.03 |
| Tensor-GaLore (64, 64, 56, 32) | 0.75 | 44.93 | 19.84 |

Table 9: Comparison of model execution times, ranks, and relative slowdown

Moreover, we have incorporated techniques such as "warm-restart" initialization of the tensor decomposition to amortize the computational overhead across training iterations. This helps minimize the impact on the overall training efficiency. We have also explored opportunities to further optimize the tensor decomposition computations, which could potentially reduce the training time slowdown even further.

**Remark 2 (Real-Valued Analysis)** *For clarity of presentation, we develop the theory of Tensor-GaLore assuming all tensors are real-valued, i.e., $\mathcal{W}_l, \mathcal{G}_t \in \mathbb{R}^{N_1 \times N_2 \times N_3 \times N_4}$ and all associated operations are in real space. This simplification allows us to focus on the core geometric and algebraic properties without the additional complexity of complex conjugates and Hermitian operations. The extension to complex-valued tensors (as needed for Fourier Neural Operators where weights may be complex in the frequency domain) is straightforward: inner products become Hermitian inner products, transposes become conjugate transposes, and orthogonality conditions incorporate complex conjugates. All main results remain valid with these natural modifications.*

## G  PARAMETER COMPLEXITY ANALYSIS

To understand the theoretical advantages of Tensor-GaLore over matrix-based GaLore, we provide a detailed analysis of the parameter complexity for both approaches. This analysis demonstrates why tensor decomposition leads to more efficient memory usage while maintaining expressiveness.

### G.1  MEMORY ANALYSIS

We provide a theoretical analysis of the memory requirements for Tensor-GaLore compared to baseline methods and matrix GaLore variants. Consider a weight tensor $W \in \mathbb{C}^{N_1 \times N_2 \times N_3 \times N_4}$ in a FNO Spectral layer. Table 10 summarizes the memory requirements for different methods. The baseline approach stores the full tensor and its corresponding optimizer states. For a rank ratio $r$ $(0 < r \leq 1)$, Tensor-GaLore requires storing the factor matrices, resulting in substantial memory savings, especially for the optimizer states. In this table, we assume the use of a complex-valued Adam optimizer, which typically requires two additional tensors (first and second moments) for each parameter.

Table 10: Theoretical memory requirements for different methods

| Method | Weight Parameters | Optimizer States (Adam) |
|---|---|---|
| Baseline | $N_1 N_2 N_3 N_4$ | $2N_1 N_2 N_3 N_4$ |
| Matrix GaLore (rollup dim 1) | $N_1 N_2 N_3 N_4$ | $2r(N_1 + N_2 N_3 N_4)$ |
| Tensor-GaLore (Tucker) | $N_1 N_2 N_3 N_4$ | $2r(N_1 + N_2 + N_3 + N_4)$ |

#### G.1.1  PROBLEM SETUP

Consider a 4D tensor weight $\mathcal{W} \in \mathbb{R}^{I_1 \times I_2 \times I_3 \times I_4}$ from a Fourier Neural Operator layer, where:

- $(I_1, I_2)$ correspond to input/output channels
- $(I_3, I_4)$ correspond to spatial frequency modes

#### G.1.2  MATRIX-BASED APPROACH (GALORE)

In the matrix-based GaLore approach, we must first reshape the tensor into a matrix. There are several possible matricization strategies:

1. $\mathbf{W}_{(1)} \in \mathbb{R}^{I_1 \times (I_2 I_3 I_4)}$
2. $\mathbf{W}_{(12)} \in \mathbb{R}^{(I_1 I_2) \times (I_3 I_4)}$

For a rank-$R$ SVD approximation of the matricized tensor:

$$\mathbf{W} \approx \mathbf{U}\mathbf{\Sigma}\mathbf{V}^H \tag{9}$$

The parameter count for storing the low-rank factors is:

- For $\mathbf{W}_{(1)}$: $R(I_1 + I_2 I_3 I_4)$ parameters
- For $\mathbf{W}_{(12)}$: $R(I_1 I_2 + I_3 I_4)$ parameters

### G.1.3  TENSOR-BASED APPROACH (TENSOR-GALORE)

In Tensor-GaLore, we use Tucker decomposition with ranks $(R_1, R_2, R_3, R_4)$:

$$\mathcal{W} \approx \mathcal{G} \times_1 \mathbf{U}^{(1)} \times_2 \mathbf{U}^{(2)} \times_3 \mathbf{U}^{(3)} \times_4 \mathbf{U}^{(4)} \tag{10}$$

where:

- $\mathcal{G} \in \mathbb{R}^{R_1 \times R_2 \times R_3 \times R_4}$ is the core tensor
- $\mathbf{U}^{(n)} \in \mathbb{R}^{I_n \times R_n}$ are the factor matrices

The total parameter count is:

$$P_{Tucker} = R_1 R_2 R_3 R_4 + \sum_{n=1}^{4} I_n R_n \tag{11}$$

### G.1.4  COMPARATIVE ANALYSIS

Let's consider a practical case where:

- $N = I_1 = I_2$ (equal input/output channels)
- $M = I_3 = I_4$ (equal spatial dimensions)
- For Tucker: $r_{max} = R_1 = R_2 = R_3 = R_4$ (equal ranks)
- For matrix SVD: $R = r_{max}^2$ (equivalent rank)

Then:

1. Matrix GaLore (best case):
$$P_{Matrix} = r_{max}^2 (N^2 + M^2) \tag{12}$$

2. Tensor-GaLore:
$$P_{Tensor} = r_{max}^4 + 2 r_{max} N + 2 r_{max} M \tag{13}$$

In typical neural operator architectures:

- $N \gg r_{max}$ (number of channels much larger than rank)
- $M \gg r_{max}$ (spatial dimensions much larger than rank)

Therefore:

- Matrix case complexity: $O(r^2(N^2 + M^2))$
- Tensor case complexity: $O(N + M + r^4)$

### G.1.5  MEMORY SAVINGS ANALYSIS

For concrete numbers, consider a typical FNO layer with:

- $N = 64$ channels
- $M = 128$ modes
- $r_{max} = 16$ (rank)

Matrix GaLore parameters:

$$P_{Matrix} = 256(64^2 + 128^2) \approx 5.2M \tag{14}$$

Tensor-GaLore parameters:

$$P_{Tensor} = 65,536 + 2(16)(64) + 2(16)(128) \approx 70K \tag{15}$$

This represents a $\sim$75x reduction in parameter count, which directly translates to memory savings in the optimizer states. The savings become even more pronounced as the spatial dimensions ($M$) increase, which is crucial for high-resolution problems.

### G.1.6 IMPACT ON EXPRESSIVENESS

Despite the significant reduction in parameters, Tensor-GaLore maintains expressiveness because:

1. The Tucker decomposition preserves the natural tensor structure of the operator

2. Each mode has its own rank parameter, allowing for more flexible approximation

3. The core tensor captures higher-order interactions between modes

This explains why Tensor-GaLore can achieve comparable or better performance while using significantly less memory than matrix-based approaches.

## H TENSOR OPERATIONS AND NOTATION

**Definition 1 (Tensor)** *An order-d tensor $\mathcal{A} \in \mathbb{R}^{I_1 \times I_2 \times \cdots \times I_d}$ is a d-dimensional array with entries $a_{i_1, i_2, \ldots, i_d}$, where $1 \leq i_k \leq I_k$ for $k = 1, \ldots, d$.*

**Definition 2 (Mode-k Unfolding)** *The mode-k unfolding of tensor $\mathcal{A}$, denoted as $\mathcal{A}_{(k)} \in \mathbb{R}^{I_k \times (I_1 \cdots I_{k-1} I_{k+1} \cdots I_d)}$, arranges the mode-k fibers as columns of the resulting matrix. Specifically:*

$$(\mathcal{A}_{(k)})_{i_k, j} = a_{i_1, \ldots, i_d}$$

*where $j = 1 + \sum_{m=1, m \neq k}^{d} (i_m - 1) \prod_{n=1, n \neq k}^{m-1} I_n$.*

**Definition 3 (Mode-k Product)** *The mode-k product of a tensor $\mathcal{A} \in \mathbb{R}^{I_1 \times \cdots \times I_d}$ with a matrix $U \in \mathbb{R}^{J \times I_k}$, denoted as $\mathcal{A} \times_k U$, results in a tensor $\mathcal{B} \in \mathbb{R}^{I_1 \times \cdots \times I_{k-1} \times J \times I_{k+1} \times \cdots \times I_d}$ with entries:*

$$(\mathcal{A} \times_k U)_{i_1, \ldots, i_{k-1}, j, i_{k+1}, \ldots, i_d} = \sum_{i_k=1}^{I_k} a_{i_1, \ldots, i_d} u_{j, i_k}$$

**Proposition 1 (Properties of Mode-k Product)** *For a tensor $\mathcal{A}$ and matrices $U, V$ of appropriate sizes:*

1. $(U \times_k \mathcal{A})_{(k)} = U \mathcal{A}_{(k)}$

2. $\mathcal{A} \times_k U \times_l V = \mathcal{A} \times_l V \times_k U$ for $k \neq l$

3. $\mathcal{A} \times_k U \times_k V = \mathcal{A} \times_k (VU)$

**Definition 4 (Tensor Inner Product)** *The inner product of two tensors $\mathcal{A}, \mathcal{B} \in \mathbb{R}^{I_1 \times \cdots \times I_d}$ is:*

$$\langle \mathcal{A}, \mathcal{B} \rangle = \sum_{i_1=1}^{I_1} \cdots \sum_{i_d=1}^{I_d} a_{i_1, \ldots, i_d} \overline{b_{i_1, \ldots, i_d}}$$

**Definition 5 (Tensor Norms)** *For a tensor $\mathcal{A}$:*

1. *Frobenius norm: $\|\mathcal{A}\|_F = \sqrt{\langle \mathcal{A}, \mathcal{A} \rangle}$*

2. *Mode-k spectral norm:* $\|\mathcal{A}\|_{(k)} = \|\mathcal{A}_{(k)}\|_2$

3. *Spectral norm:* $\|\mathcal{A}\| = \max_{\|x^{(k)}\|=1} \|\mathcal{A} \times_1 x^{(1)} \times_2 \cdots \times_d x^{(d)}\|$

**Definition 6 (Tensor Outer Product)** *The outer product of vectors $u^{(k)} \in \mathbb{R}^{I_k}$ for $k = 1, \ldots, d$ is a tensor $\mathcal{A} = u^{(1)} \circ u^{(2)} \circ \cdots \circ u^{(d)}$ with entries:*

$$a_{i_1,\ldots,i_d} = u_{i_1}^{(1)} u_{i_2}^{(2)} \cdots u_{i_d}^{(d)}$$

**Definition 7 (Tensor Contraction)** *The contraction of a tensor $\mathcal{A} \in \mathbb{R}^{I_1 \times \cdots \times I_d}$ along modes $p$ and $q$ (where $I_p = I_q$) is:*

$$(Contract_{p,q}(\mathcal{A}))_{i_1,\ldots,i_{p-1},i_{p+1},\ldots,i_{q-1},i_{q+1},\ldots,i_d} = \sum_{i=1}^{I_p} a_{i_1,\ldots,i_{p-1},i,i_{p+1},\ldots,i_{q-1},i,i_{q+1},\ldots,i_d}$$

### H.1 TENSOR TRACE AND INNER PRODUCTS

**Definition 8 (Tensor Inner Product)** *For tensors $\mathcal{A}, \mathcal{B} \in \mathbb{R}^{I_1 \times I_2 \times \cdots \times I_d}$, their inner product is:*

$$\langle \mathcal{A}, \mathcal{B} \rangle = \sum_{i_1=1}^{I_1} \sum_{i_2=1}^{I_2} \cdots \sum_{i_d=1}^{I_d} \mathcal{A}_{i_1,i_2,\ldots,i_d} \mathcal{B}_{i_1,i_2,\ldots,i_d}$$

**Definition 9 (Tensor Trace)** *For a tensor $\mathcal{A}$, there are several equivalent ways to understand its trace:*

*1. Mode-wise trace:*

$$tr_k(\mathcal{A}) = \sum_{i_k=1}^{I_k} \mathcal{A}_{i_1,\ldots,i_k,\ldots,i_d}|_{i_k=i_k}$$

*2. Using mode-k unfolding:*

$$tr(\mathcal{A}_{(k)}) = \sum_{i=1}^{I_k} (\mathcal{A}_{(k)})_{i,i}$$

*3. Inner product interpretation: When used in expressions like $tr(d\mathcal{W}_l^\top \times_1 X \times_2 Y)$, this is actually computing:*

$$\langle d\mathcal{W}_l, X \otimes Y \rangle$$

**Proposition 2 (Key Properties)** *For the trace operation in tensor gradients:*

*1. Inner Product Form:*

$$tr(d\mathcal{W}^\top \times_1 X \times_2 Y) = \langle d\mathcal{W}, X \otimes Y \rangle$$

*2. Differential Form: For scalar function $\phi$ and tensor $\mathcal{W}$:*

$$d\phi = tr(d\mathcal{W}^\top \times_1 X \times_2 Y) \implies \frac{\partial \phi}{\partial \mathcal{W}} = X \otimes Y$$

*3. Mode-wise Consistency:*

$$tr(d\mathcal{W}^\top \times_1 X \times_2 Y) = tr(X^\top d\mathcal{W}_{(1)} Y)$$

*where $d\mathcal{W}_{(1)}$ is the mode-1 unfolding.*

**Example 1** *In the logsoftmax gradient computation:*

$$-d\phi = tr(d\mathcal{W}_l^\top \times_1 (P_1^\perp y)^\top \mathcal{J}_l \times_2 f_{l-1}^\top)$$
$$= \langle d\mathcal{W}_l, \mathcal{J}_l^\top P_1^\perp y \otimes f_{l-1} \rangle$$

*This leads to the gradient term:*

$$\mathcal{G}_l = \mathcal{J}_l^\top P_1^\perp y \otimes f_{l-1}$$

**Remark 3 (Connection to Matrix Case)** *When working with matrices, the trace operation reduces to the familiar form:*

$$tr(A^\top B) = \langle A, B \rangle = \sum_{i,j} A_{ij} B_{ij}$$

*The tensor trace generalizes this to handle higher-order tensors while preserving the key property that it relates to directional derivatives through inner products.*

## H.2 STABLE RANK FOR TENSORS

**Definition 10 (Matrix Stable Rank)** *For a matrix A, the stable rank is defined as:*

$$sr(A) := \frac{\|A\|_F^2}{\|A\|_2^2}$$

*where $\|\cdot\|_F$ is the Frobenius norm and $\|\cdot\|_2$ is the spectral norm.*

**Definition 11 (Tensor Stable Rank)** *For a non-zero tensor $\mathcal{T} \in \mathbb{R}^{N_1 \times N_2 \times \ldots \times N_d}$, we define the mode-wise stable rank vector as:*

$$sr(\mathcal{T}) = [sr_1(\mathcal{T}), sr_2(\mathcal{T}), \ldots, sr_d(\mathcal{T})]$$

*where for each mode $k$:*

$$sr_k(\mathcal{T}) := \frac{\|\mathcal{T}\|_F^2}{\|\mathcal{T}_{(k)}\|_2^2}$$

*Here:*

- $\mathcal{T}_{(k)}$ *is the mode-k unfolding of tensor $\mathcal{T}$*

- $\|\mathcal{T}\|_F^2 = \sum_{i_1,\ldots,i_d} |\mathcal{T}_{i_1,\ldots,i_d}|^2$ *is the tensor Frobenius norm*

- $\|\mathcal{T}_{(k)}\|_2$ *is the spectral norm of the mode-k unfolding*

**Lemma 1 (Tensor-Matrix Norm Relations)** *For any tensor $\mathcal{T}$ and its mode-k unfolding $\mathcal{T}_{(k)}$:*

$$\|\mathcal{T}\|_F = \|\mathcal{T}_{(k)}\|_F$$

*This follows from the fact that unfolding is just a rearrangement of entries.*

**Proposition 3 (Properties of Tensor Stable Rank)** *For a non-zero tensor $\mathcal{T}$:*

1. *Each $sr_k(\mathcal{T}) \geq 1$*

2. *$sr_k(\mathcal{T}) \leq rank(\mathcal{T}_{(k)})$*

3. *$sr_k(\mathcal{T})$ is invariant under orthogonal transformations in mode $k$*

4. *For a rank-1 tensor, $sr_k(\mathcal{T}) = 1$ for all $k$*

**Proof 1** *1. For any matrix $M$, we know $\|M\|_F^2 \geq \|M\|_2^2$. Therefore:*

$$sr_k(\mathcal{T}) = \frac{\|\mathcal{T}\|_F^2}{\|\mathcal{T}_{(k)}\|_2^2} = \frac{\|\mathcal{T}_{(k)}\|_F^2}{\|\mathcal{T}_{(k)}\|_2^2} \geq 1$$

*where we used the tensor-matrix norm relation lemma.*

*2. For any matrix $M$ of rank $r$:*

$$\|M\|_2^2 \geq \frac{\|M\|_F^2}{r}$$

*Applying this to $\mathcal{T}_{(k)}$:*

$$sr_k(\mathcal{T}) = \frac{\|\mathcal{T}_{(k)}\|_F^2}{\|\mathcal{T}_{(k)}\|_2^2} \leq rank(\mathcal{T}_{(k)})$$

3. *For any orthogonal transformation $U$ in mode $k$:*

$$\|U\mathcal{T}_{(k)}\|_F = \|\mathcal{T}_{(k)}\|_F \text{ and } \|U\mathcal{T}_{(k)}\|_2 = \|\mathcal{T}_{(k)}\|_2$$

4. *For a rank-1 tensor $\mathcal{T} = a_1 \otimes ... \otimes a_d$:*

- *Each mode-k unfolding is rank-1*

- *For rank-1 matrices, $\|M\|_F^2 = \|M\|_2^2$*

- *Therefore $sr_k(\mathcal{T}) = 1$*

**Definition 12 (Multilinear Stable Rank)** *For a tensor $\mathcal{T}$, the multilinear stable rank is:*

$$msr(\mathcal{T}) := \min_k sr_k(\mathcal{T})$$

*This provides a lower bound on the minimal mode-k rank needed to approximate $\mathcal{T}$.*

**Remark 4 (Connection to Low-Rank Approximation)** *The stable rank of a tensor in each mode provides insight into how well it can be approximated by a low-rank decomposition:*

*1. If $sr_k(\mathcal{T})$ is close to 1 in mode $k$, then $\mathcal{T}$ is nearly low-rank in that mode*

*2. For a Tucker decomposition:*

$$\mathcal{T} \approx \mathcal{G} \times_1 U^{(1)} \times_2 U^{(2)}... \times_d U^{(d)}$$

*The stable rank helps determine appropriate ranks for each mode*

**Remark 5 (Application to FNO)** *For FNO weight tensors $\mathcal{R} \in \mathbb{R}^{N_1 \times N_2 \times N_3 \times N_4}$:*

*1. Mode-1 and Mode-2 typically correspond to input/output channels 2. Mode-3 and Mode-4 correspond to Fourier modes 3. Stable rank in Fourier modes often naturally decreases due to spectral decay*

H.3    POSITIVE SEMI-DEFINITENESS FOR TENSORS

**Definition 13 (Mode-k PSD Tensor)** *A tensor $\mathcal{T} \in \mathbb{R}^{N_1 \times N_2 \times \cdots \times N_d}$ is called mode-k positive semi-definite if its mode-k unfolding $\mathcal{T}_{(k)} \in \mathbb{R}^{N_k \times (N_1 \cdots N_{k-1} N_{k+1} \cdots N_d)}$ satisfies:*

$$x^\top \mathcal{T}_{(k)} x \geq 0 \quad \forall x \in \mathbb{R}^{N_k}$$

**Definition 14 (All-modes PSD Tensor)** *A tensor $\mathcal{T}$ is called all-modes positive semi-definite if it is mode-k PSD for all modes $k$.*

**Definition 15 (Strong PSD Tensor)** *A tensor $\mathcal{T} \in \mathbb{R}^{N_1 \times N_2 \times \cdots \times N_d}$ is called strongly positive semi-definite if:*

$$\mathcal{T} \times_1 x_1 \times_2 x_2 \times_3 \cdots \times_d x_d \geq 0$$

*for all vectors $x_k \in \mathbb{R}^{N_k}, k = 1, \ldots, d$.*

**Lemma 2 (Hierarchy of PSD Definitions)** *For a tensor $\mathcal{T}$:*

$$\text{Strong PSD} \implies \text{All-modes PSD} \implies \text{Mode-k PSD}$$

*The reverse implications do not necessarily hold.*

**Remark 6 (For Tensor-GaLore)** *For our generalized gradient analysis, we propose to use:*

*1. Mode-specific PSD condition:*

$$\mathcal{B}_i \text{ and } \mathcal{C}_i \text{ are mode-k PSD for relevant modes } k$$

*2. This means for each mode $k$:*

- $(\mathcal{B}_i)_{(k)}$ *is a PSD matrix*

- $(\mathcal{C}_i)_{(k)}$ *is a PSD matrix*

- *The tensor operator* $\mathcal{S}_k = \frac{1}{N} \sum_{i=1}^{N} \mathcal{C}_i \otimes_k \mathcal{B}_i$ *is well-defined*

*3. This ensures:*

- *The mode-k eigenvalues* $\lambda_1^{(k)}, \lambda_2^{(k)}$ *are real and non-negative*

- *The projection onto minimal eigenspace is well-defined for each mode*

- *The stable rank bounds make sense mode-wise*

**Proposition 4 (For FNO)** *In FNO, the tensors $\mathcal{B}_i$ and $\mathcal{C}_i$ naturally satisfy mode-k PSD conditions because:*

*1. For channel modes (1,2):*

- *Unfoldings correspond to standard channel operations*

- *PSD property follows from network structure*

*2. For Fourier modes (3,4):*

- *Unfoldings correspond to frequency domain operations*

- *PSD property follows from spectral properties*

**Corollary 1 (Implications for Gradient Analysis)** *The mode-k PSD property ensures:*

*1. Each mode has real, non-negative eigenvalues:*

$$0 \leq \lambda_1^{(k)} < \lambda_2^{(k)} \leq \cdots$$

*2. Mode-wise stable rank bounds are well-defined:*

$$sr_k(\mathcal{G}_t) \leq sr_k(\mathcal{G}_{t_0}^{\parallel}) + \text{decay term}$$

*3. The gradient naturally becomes low-rank in each mode independently.*

***Definition 16 (Lipschitz Continuity)*** *A function $h : \mathcal{X} \rightarrow \mathcal{Y}$ between normed spaces has L-continuity (is L-Lipschitz) if for any $x_1, x_2 \in \mathcal{X}$:*

$$\|h(x_1) - h(x_2)\|_{\mathcal{Y}} \leq L\|x_1 - x_2\|_{\mathcal{X}}$$

*For tensors, this generalizes to mode-wise continuity:*

- *Matrix case ($d = 2$): Standard Lipschitz continuity with Frobenius norm*

- *Tensor case ($d > 2$): Mode-k Lipschitz continuity for each mode k*

- *Neural networks: Composition of Lipschitz continuous operations*

# I    REVERSIBILITY OF FOURIER NEURAL OPERATORS

## I.1    DEFINITION AND PRELIMINARIES

**Definition 17 (Reversibility)** *A network $\mathcal{N}$ that maps input $x$ to output $y = \mathcal{N}(x)$ is reversible if there exists $J(x)$ such that:*

1. *Forward: $y = J(x)x$*

2. *Backward: $dx = J(x)^{\top} dy$*

*where $J(x)$ can be a function of both input and weights.*

## I.2 SPECTRAL LAYER

**Lemma 3 (Spectral Layer Reversibility)** *The FNO spectral convolution layer $(Kv)(x) = \mathcal{F}^{-1}(R \cdot \mathcal{F}v)(x)$ is reversible, where $R$ is the learnable weight tensor in Fourier space.*

The spectral layer consists of three operations:

1. Fourier transform: $\mathcal{F} : v \mapsto \hat{v}$

2. Linear transform in Fourier space: $R\cdot : \hat{v} \mapsto R\hat{v}$

3. Inverse Fourier: $\mathcal{F}^{-1} : R\hat{v} \mapsto \mathcal{F}^{-1}(R\hat{v})$

We can express the complete operation as:

$$Kv = J_K(x)v \text{ where } J_K(x) = \mathcal{F}^{-1}R\mathcal{F}$$

For the backward pass:

$$dv = J_K(x)^\top dy = \mathcal{F}^\top R^\top (\mathcal{F}^{-1})^\top dy$$

Since $\mathcal{F}$ is unitary: $\mathcal{F}^\top = \mathcal{F}^{-1}$ and $(\mathcal{F}^{-1})^\top = \mathcal{F}$, we have:

$$dv = \mathcal{F}^{-1}R^\top \mathcal{F}dy$$

Therefore:

- Forward pass: $y = J_K(x)x$
- Backward pass: $dx = J_K(x)^\top dy$

Thus satisfying the reversibility conditions, regardless of the size or rank of $R$.

## I.3 MLP LAYER

**Lemma 4 (MLP Layer Reversibility)** *The MLP layer with weight matrix $W$ mapping $v \mapsto Wv$ is reversible.*

1. Forward pass: $y = Wv$

2. Set $J_W(x) = W$

3. Backward pass: $dv = W^\top dy = J_W(x)^\top dy$

The linear layer satisfies reversibility conditions directly, even when $W$ is rank-deficient.

## I.4 ACTIVATION FUNCTION

**Lemma 5 (Activation Reversibility)** *If the activation function $\sigma$ is reversible (e.g., LeakyReLU), then its application is reversible.*

Consider LeakyReLU with parameter $0 < a < 1$:

1. Forward: $y = \max(ax, x)$

2. Set $J_\sigma(x) = \text{diag}(\mathbf{1}[x > 0] + a \cdot \mathbf{1}[x \leq 0])$

3. Backward: $dx = J_\sigma(x)^\top dy$

This matches the required reversibility form.

## I.5 FULL FNO ANALYSIS

**Lemma 6 (FNO Block Reversibility)** *An FNO block consisting of spectral layer $(K)$, MLP layer $(W)$, and reversible activation $(\sigma)$ is reversible.*

Let $N = (\sigma \circ W \circ K)$ be an FNO block.

From previous theorems, we have:

- Spectral layer: $v \mapsto J_K(x)v = \mathcal{F}^{-1}(R\mathcal{F}v)$
- MLP layer: $v \mapsto J_W(x)v = Wv$
- Activation: $v \mapsto J_\sigma(x)v$

By composition:
$$y = J_{\text{block}}(x)v$$
where $J_{\text{block}}(x) = J_\sigma(x)J_W(x)J_K(x)$

For backward pass:
$$dv = J_K(x)^\top J_W(x)^\top J_\sigma(x)^\top dy = J_{\text{block}}(x)^\top dy$$

Therefore, the full block is reversible.

**Lemma 7 (Full FNO Reversibility)** *A full FNO network with reversible activations is reversible.*

Consider a full FNO with blocks $N_1, N_2, ..., N_L$:

1. Each block $N_i$ has its $J_i(x)$ from previous theorem
2. By sequential composition:
$$y = J_{\text{FNO}}(x)v$$
   where $J_{\text{FNO}}(x) = J_L(x)J_{L-1}(x)...J_1(x)$
3. The backward pass follows from composition:
$$dv = J_1(x)^\top ... J_{L-1}(x)^\top J_L(x)^\top dy = J_{\text{FNO}}(x)^\top dy$$

Therefore, the full FNO with reversible activations satisfies the reversibility conditions.

**Lemma 8 (Gradient Form for Tensor Reversible Models)** *Consider a chained reversible neural network $\mathcal{N}(x) := \mathcal{N}_L(\mathcal{N}_{L-1}(...\mathcal{N}_1(x)))$ and define:*

- $\mathcal{J}_l := Jacobian(\mathcal{N}_L)...Jacobian(\mathcal{N}_{l+1})$
- $f_l := \mathcal{N}_l(...\mathcal{N}_1(x))$

*Then the weight tensor $\mathcal{W}_l \in \mathbb{R}^{N_1 \times N_2 \times N_3 \times N_4}$ at layer $l$ has gradient $\mathcal{G}_l$ in the following form for batch size 1:*

*(a) For $\ell_2$-objective $\phi := \frac{1}{2}\|y - f_L\|_2^2$:*
$$\mathcal{G}_l = \mathcal{J}_l^\top y \otimes f_{l-1} - (\mathcal{J}_l^\top \mathcal{J}_l \mathcal{W}_l \times_1 f_{l-1}) \otimes f_{l-1}$$

*(b) For $K$-way logsoftmax loss $\phi(y; f_L) := -\log\left(\frac{\exp(y^\top f_L)}{\mathbf{1}^\top \exp(f_L)}\right)$ with small logits $\|P_1^\perp f_L\|_\infty \ll \sqrt{K}$:*
$$\mathcal{G}_l = (\mathcal{J}_l P_1^\perp y - \gamma K^{-1} \mathcal{J}_l^\top P_1^\perp \mathcal{J}_l \mathcal{W}_l \times_1 f_{l-1}) \otimes f_{l-1}$$

*where:*

- $\gamma \approx 1$
- $y$ *is a data label with* $y^\top \mathbf{1} = 1$

- $P_1^\perp := I - \frac{1}{K}\mathbf{1}\mathbf{1}^\top$ is the zero-mean PSD projection matrix

- $\times_k$ denotes mode-k tensor product

- $\otimes$ denotes tensor outer product

**Proof 2** *Note that for layered reversible network, we have*

$$\mathcal{N}(x) = \mathcal{N}_L(\mathcal{N}_{L-1}(...\mathcal{N}_1(x))) = \mathcal{K}_L(x)\mathcal{K}_{L-1}(x)...\mathcal{K}_1(x)x$$

*Let $f_l := \mathcal{N}_l(\mathcal{N}_{l-1}(...\mathcal{N}_1(x)))$ and $\mathcal{J}_l := \mathcal{K}_L(x)...\mathcal{K}_{l+1}(x)$, and for linear layer $l$, we can write $\mathcal{N}(x) = \mathcal{J}_l \times_1 (\mathcal{W}_l \times_1 f_{l-1})$. Therefore, for the linear layer $l$ with weight tensor $\mathcal{W}_l$, we have:*

$$
\begin{aligned}
d\phi &= (y - \mathcal{N}(x))^\top d\mathcal{N}(x)\\
&= (y - \mathcal{N}(x))^\top (\mathcal{K}_L(x)...\mathcal{K}_{l+1}(x))(d\mathcal{W}_l \times_1 f_{l-1}) + \text{ terms not related to } d\mathcal{W}_l\\
&= (y - \mathcal{J}_l \times_1 (\mathcal{W}_l \times_1 f_{l-1}))^\top \mathcal{J}_l \times_1 (d\mathcal{W}_l \times_1 f_{l-1})\\
&= tr(d\mathcal{W}_l^\top \times_1 (\mathcal{J}_l^\top (y - \mathcal{J}_l \times_1 (\mathcal{W}_l \times_1 f_{l-1}))) \times_2 f_{l-1}^\top)
\end{aligned}
$$

*This gives the gradient of $\mathcal{W}_l$:*

$$\mathcal{G}_l = \mathcal{J}_l^\top y \otimes f_{l-1} - (\mathcal{J}_l^\top \mathcal{J}_l \times_1 (\mathcal{W}_l \times_1 f_{l-1})) \otimes f_{l-1}$$

*where:*

- $\times_k$ denotes the mode-k product between a tensor and a matrix

- $\otimes$ denotes the tensor outer product

- The gradient $\mathcal{G}_l$ has the same dimensionality as $\mathcal{W}_l$

**Remark 7 (Gradient Form for Tensor Reversible Models with Dimensions)** *Consider a chained reversible neural network $\mathcal{N}(x)$ where: Input $x \in \mathbb{R}^M$, Output $y \in \mathbb{R}^K$, Weight tensor $\mathcal{W}_l \in \mathbb{R}^{N_1 \times N_2 \times N_3 \times N_4}$, Layer output $f_l \in \mathbb{R}^{N_l}$ and Jacobian $\mathcal{J}_l \in \mathbb{R}^{K \times N_l}$.*

*Then for batch size 1: (a) For $\ell_2$-objective $\phi := \frac{1}{2}\|y - f_L\|_2^2$:*

$$\mathcal{G}_l = \mathcal{J}_l^\top y \otimes f_{l-1} - (\mathcal{J}_l^\top \mathcal{J}_l \mathcal{W}_l \times_1 f_{l-1}) \otimes f_{l-1}$$

*where $\mathcal{J}_l^\top y \in \mathbb{R}^{N_l}$, $f_{l-1} \in \mathbb{R}^{N_{l-1}}$ and the final gradient $\mathcal{G}_l \in \mathbb{R}^{N_1 \times N_2 \times N_3 \times N_4}$.*

**Proof 3** *1) Let us start with the initial setup:*

$$\mathcal{N}(x) = \mathcal{K}_L(x)\mathcal{K}_{L-1}(x)...\mathcal{K}_1(x)x$$

*where each $\mathcal{K}_i$ maps $\mathbb{R}^{N_{i-1}} \to \mathbb{R}^{N_i}$*

*2) For linear layer $l$:*

- $f_{l-1} \in \mathbb{R}^{N_{l-1}}$ is input

- $\mathcal{W}_l \in \mathbb{R}^{N_1 \times N_2 \times N_3 \times N_4}$ is weight tensor

- $\mathcal{W}_l \times_1 f_{l-1}$ maps to $\mathbb{R}^{N_l}$

- $\mathcal{J}_l \in \mathbb{R}^{K \times N_l}$ is Jacobian

*3) Then, like before we do the differential computation:*

$$
\begin{aligned}
d\phi &= (y - \mathcal{N}(x))^\top d\mathcal{N}(x) && [\mathbb{R}^K \times \mathbb{R}^K \to \mathbb{R}]\\
&= (y - \mathcal{N}(x))^\top \mathcal{J}_l(d\mathcal{W}_l \times_1 f_{l-1}) && [\mathbb{R}^K \times \mathbb{R}^{K \times N_l} \times \mathbb{R}^{N_l} \to \mathbb{R}]\\
&= (y - \mathcal{J}_l \times_1 (\mathcal{W}_l \times_1 f_{l-1}))^\top \mathcal{J}_l \times_1 (d\mathcal{W}_l \times_1 f_{l-1})
\end{aligned}
$$

*4) Mode-wise analysis for gradient:*

- **First term:** $\mathcal{J}_l^\top y \otimes f_{l-1}$ - $\mathcal{J}_l^\top y \in \mathbb{R}^{N_l}$ - $f_{l-1} \in \mathbb{R}^{N_{l-1}}$ - *Outer product gives tensor in* $\mathbb{R}^{N_1 \times N_2 \times N_3 \times N_4}$

- **Second term:** $(\mathcal{J}_l^\top \mathcal{J}_l \mathcal{W}_l \times_1 f_{l-1}) \otimes f_{l-1}$ - $\mathcal{J}_l^\top \mathcal{J}_l \in \mathbb{R}^{N_l \times N_l}$ - $\mathcal{W}_l \times_1 f_{l-1} \in \mathbb{R}^{N_l}$ - *Result is tensor in* $\mathbb{R}^{N_1 \times N_2 \times N_3 \times N_4}$

*5) Therefore final gradient:*

$$\mathcal{G}_l = \mathcal{J}_l^\top y \otimes f_{l-1} - (\mathcal{J}_l^\top \mathcal{J}_l \mathcal{W}_l \times_1 f_{l-1}) \otimes f_{l-1} \in \mathbb{R}^{N_1 \times N_2 \times N_3 \times N_4}$$

*We finally have a gradient tensor of the same shape as* $\mathcal{W}_l$.

**Remark 8** *We only wanted to show an example of checking all the dimensions to ensure they match the generalized version for tensors. In the following subsequent proofs and lemma, we don't keep track of it all, but we give appropriate dimensions wherever necessary.*

**Lemma 9 (Tensor Gradient Form for Logsoftmax)** *For a reversible network with weight tensor* $\mathcal{W}_l$ *at layer* $l$, *under the* $K$-*way logsoftmax loss with small logits, the gradient has the form:*

$$\mathcal{G}_l = (\mathcal{J}_l \times_1 P_1^\perp y - \gamma K^{-1} \mathcal{J}_l^\top \times_1 P_1^\perp \times_2 \mathcal{J}_l \times_1 (\mathcal{W}_l \times_1 f_{l-1})) \otimes f_{l-1}$$

**Proof 4** *Starting with the differential form above:*

*1. For reversible network,* $d\mathcal{N}(x) = \mathcal{J}_l \times_1 (d\mathcal{W}_l \times_1 f_{l-1})$

*2. The zero-mean projection in the tensor form:*

$$d\hat{f} = P_1^\perp d\mathcal{N}(x)$$
$$= P_1^\perp \mathcal{J}_l \times_1 (d\mathcal{W}_l \times_1 f_{l-1})$$

*3. Substituting into the logsoftmax differential:*

$$-d\phi = y^\top P_1^\perp \mathcal{J}_l \times_1 (d\mathcal{W}_l \times_1 f_{l-1})$$
$$- \gamma K^{-1} \hat{f}^\top P_1^\perp \mathcal{J}_l \times_1 (d\mathcal{W}_l \times_1 f_{l-1})$$
$$+ O(\hat{f}^2/K) \ terms$$

*4. Under small logits assumption, the* $O(\hat{f}^2/K)$ *terms become negligible*

*5. Express in tensor form:*

$$-d\phi = tr(d\mathcal{W}_l^\top \times_1 (P_1^\perp y)^\top \mathcal{J}_l \times_2 f_{l-1}^\top)$$
$$- \gamma K^{-1} tr(d\mathcal{W}_l^\top \times_1 (P_1^\perp \mathcal{J}_l \times_1 (\mathcal{W}_l \times_1 f_{l-1}))^\top \mathcal{J}_l \times_2 f_{l-1}^\top)$$

*6. Therefore, the gradient is:*

$$\mathcal{G}_l = (\mathcal{J}_l \times_1 P_1^\perp y - \gamma K^{-1} \mathcal{J}_l^\top \times_1 P_1^\perp \times_2 \mathcal{J}_l \times_1 (\mathcal{W}_l \times_1 f_{l-1})) \otimes f_{l-1}$$

# J THEORETICAL RESULTS OF TENSOR-GALORE FOR NEURAL OPERATORS

**Lemma 10 (Tensor Gradient becomes low-rank during training)** *Suppose the gradient tensor follows the parametric form:*

$$\mathcal{G}_t = \frac{1}{N} \sum_{i=1}^N (\mathcal{A}_i - \mathcal{B}_i \times_1 \mathcal{W}_t \times_2 \mathcal{C}_i)$$

*with constant* $\mathcal{A}_i \in \mathbb{R}^{N_1 \times N_2 \times N_3 \times N_4}$, *PSD tensors* $\mathcal{B}_i$ *and* $\mathcal{C}_i$ *after* $t \geq t_0$. *We study vanilla SGD weight update:* $\mathcal{W}_t = \mathcal{W}_{t-1} + \eta \mathcal{G}_{t-1}$.

*Let* $\mathcal{S}_k := \frac{1}{N} \sum_{i=1}^N \mathcal{C}_i \otimes_k \mathcal{B}_i$ *be the mode-*$k$ *tensor operator and* $\lambda_1^{(k)} < \lambda_2^{(k)}$ *its two smallest distinct eigenvalues for each mode* $k$. *Then the mode-wise stable rank satisfies:*

$$sr_k(\mathcal{G}_t) \leq sr_k(\mathcal{G}_{t_0}^{\|}) + \left(\frac{1 - \eta\lambda_2^{(k)}}{1 - \eta\lambda_1^{(k)}}\right)^{2(t-t_0)} \frac{\|\mathcal{G}_0 - \mathcal{G}_{t_0}^{\|}\|_F^2}{\|\mathcal{G}_{t_0}^{\|}\|_2^2}$$

*where:*

- *$sr_k(\mathcal{G}_t)$ is the mode-k stable rank of gradient tensor at time t*

- *$\mathcal{G}_{t_0}^{\|}$ is the projection of $\mathcal{G}_{t_0}$ onto the minimal eigenspace $\mathcal{V}_1^{(k)}$ of $\mathcal{S}_k$ corresponding to $\lambda_1^{(k)}$ for each mode k*

- *$\|\cdot\|_F$ is the tensor Frobenius norm*

- *$\|\cdot\|_2$ is the spectral norm of the mode-k unfolding*

- *$\times_k$ denotes mode-k tensor product*

*Furthermore, the multilinear stable rank satisfies:*

$$msr(\mathcal{G}_t) \leq \min_k \left\{ sr_k(\mathcal{G}_{t_0}^{\|}) + \left(\frac{1 - \eta\lambda_2^{(k)}}{1 - \eta\lambda_1^{(k)}}\right)^{2(t-t_0)} \frac{\|\mathcal{G}_0 - \mathcal{G}_{t_0}^{\|}\|_F^2}{\|\mathcal{G}_{t_0}^{\|}\|_2^2} \right\}$$

**Proof 5** *1) First, we derive the recursive update rule for the gradient tensor. We have:*

$$\mathcal{G}_t = \frac{1}{N}\sum_{i=1}^{N}(\mathcal{A}_i - \mathcal{B}_i \times_1 \mathcal{W}_t \times_2 \mathcal{C}_i)$$

$$= \frac{1}{N}\sum_{i=1}^{N}(\mathcal{A}_i - \mathcal{B}_i \times_1 (\mathcal{W}_{t-1} + \eta\mathcal{G}_{t-1}) \times_2 \mathcal{C}_i)$$

$$= \frac{1}{N}\sum_{i=1}^{N}\mathcal{A}_i - \frac{1}{N}\sum_{i=1}^{N}\mathcal{B}_i \times_1 \mathcal{W}_{t-1} \times_2 \mathcal{C}_i - \eta\frac{1}{N}\sum_{i=1}^{N}\mathcal{B}_i \times_1 \mathcal{G}_{t-1} \times_2 \mathcal{C}_i$$

$$= \mathcal{G}_{t-1} - \eta\frac{1}{N}\sum_{i=1}^{N}\mathcal{B}_i \times_1 \mathcal{G}_{t-1} \times_2 \mathcal{C}_i$$

*2) For each mode k, let's consider the mode-k unfolding. Define the tensor operator:*

$$\mathcal{S}_k := \frac{1}{N}\sum_{i=1}^{N}\mathcal{C}_i \otimes_k \mathcal{B}_i$$

*Then for the mode-k unfolding $(\mathcal{G}_t)_{(k)}$:*

$$(\mathcal{G}_t)_{(k)} = (\mathcal{G}_{t-1})_{(k)} - \eta\mathcal{S}_k(\mathcal{G}_{t-1})_{(k)} \tag{16}$$

*3) Since $\mathcal{B}_i$ and $\mathcal{C}_i$ are mode-k PSD, $\mathcal{S}_k$ is a PSD operator. Let $\lambda_1^{(k)} < \lambda_2^{(k)}$ be its two smallest distinct eigenvalues. Let $\mathcal{V}_1^{(k)}$ be the eigenspace corresponding to $\lambda_1^{(k)}$.*

*4) For any mode k, we can decompose $(\mathcal{G}_{t_0})_{(k)}$ into parallel and perpendicular components:*

$$(\mathcal{G}_{t_0})_{(k)} = (\mathcal{G}_{t_0}^{\|})_{(k)} + (\mathcal{G}_{t_0}^{\perp})_{(k)}$$

*where $(\mathcal{G}_{t_0}^{\|})_{(k)}$ is the projection onto $\mathcal{V}_1^{(k)}$.*

*5) The mode-k unfolded gradient follows:*

$$(\mathcal{G}_t)_{(k)} = (I - \eta\mathcal{S}_k)^{t-t_0}(\mathcal{G}_{t_0})_{(k)}$$

*6) Using the spectral properties of $\mathcal{S}_k$:*

$$\|(\mathcal{G}_t)_{(k)}\|_F^2 \le (1 - \eta\lambda_2^{(k)})^{2(t-t_0)}\|(\mathcal{G}_{t_0}^{\perp})_{(k)}\|_F^2 + (1 - \eta\lambda_1^{(k)})^{2(t-t_0)}\|(\mathcal{G}_{t_0}^{\parallel})_{(k)}\|_F^2$$

*7) For the mode-k stable rank:*

$$sr_k(\mathcal{G}_t) = \frac{\|(\mathcal{G}_t)_{(k)}\|_F^2}{\|(\mathcal{G}_t)_{(k)}\|_2^2}$$

$$\le sr_k(\mathcal{G}_{t_0}^{\parallel}) + \left(\frac{1 - \eta\lambda_2^{(k)}}{1 - \eta\lambda_1^{(k)}}\right)^{2(t-t_0)} \frac{\|\mathcal{G}_0 - \mathcal{G}_{t_0}^{\parallel}\|_F^2}{\|\mathcal{G}_{t_0}^{\parallel}\|_2^2}$$

*8) Finally, for the multilinear stable rank:*

$$msr(\mathcal{G}_t) = \min_k sr_k(\mathcal{G}_t)$$

*Therefore, the bound holds for each mode independently.*

**Remark 9** *For FNO specifically:*

1. *Fourier modes (3,4) may have different stable rank behavior than channel modes (1,2)*

2. *Natural frequency decay affects eigenvalue structure in Fourier modes*

3. *Channel modes might maintain higher stable rank due to information preservation needs*

4. *Overall low-rank structure emerges from combined effect across all modes*

**Corollary 2 (Low-rank Tensor Gradient)** *If the gradient takes the parametric form*

$$\mathcal{G}_t = \frac{1}{N} \sum_{i=1}^{N} (\mathcal{A}_i - \mathcal{B}_i \times_1 \mathcal{W}_t \times_2 f_i) \otimes f_i$$

*with all $\mathcal{B}_i$ mode-k full-rank, and $N' := rank(\{f_i\}) < n$, then for each mode $k$:*

$$sr_k(\mathcal{G}_{t_0}^{\parallel}) \le n_k - N'$$

*and thus $sr_k(\mathcal{G}_t) \le n_k/2$ for large t, where $n_k$ is the dimension of mode k.*

**Proof 6** *Similar to the Galore paper, it's easy to analyze mode by mode.*

*1) Let $\mathcal{C}_i = f_i \otimes f_i^{\top}$. Since $N' := rank(\{f_i\}_{i=1}^{N}) < n$ and $f_i \in \mathbb{R}^n$, the collections of vectors $\{f_i\}_{i=1}^{N}$ cannot span the entire space $\mathbb{R}^n$.*

*2) For each mode k:*

- *Let $\{u_j\}_{j=1}^{n-N'}$ be orthonormal bases for the null space of $\{f_i\}_{i=1}^{N}$*

- *Let $\{e_k\}_{k=1}^{n_k}$ be orthonormal bases for $\mathbb{R}^{n_k}$*

- *The product bases $\{u_j \otimes e_k\}$ form a set of bases for the minimal eigenspace $\mathcal{V}_1^{(k)}$ of $\mathcal{S}_k$ with minimal eigenvalue 0*

- *Since $\mathcal{B}_i$ are mode-k full-rank, no extra dimensions exist for $\mathcal{V}_1^{(k)}$*

*3) For the mode-k projection of $\mathcal{G}_{t_0}$ onto $\mathcal{V}_1^{(k)}$:*

$$(\mathcal{G}_{t_0}^{\parallel})_{(k)} = \sum_{j=1}^{n-N'} \sum_{l=1}^{n_k} c_{jl} u_j e_l^{\top} = \sum_{j=1}^{n-N'} u_j \left(\sum_{l=1}^{n_k} c_{jl} e_l\right)^{\top}$$

*4) Therefore:*

$$sr_k(\mathcal{G}_{t_0}^{\parallel}) \le rank((\mathcal{G}_{t_0}^{\parallel})_{(k)}) \le n_k - N'$$

*since stable rank is a lower-bound of the rank in each mode.*

*5) On the other hand, $\mathcal{G}_t$ can be written as a summation of $N'$ rank-1 tensors by representing each $f_i = \sum_{j=1}^{N'} b_{ij} f_j'$ as a linear combination of $\{f_j'\}_{j=1}^{N'}$:*

$$\mathcal{G}_t = \frac{1}{N} \sum_{i=1}^{N} (\mathcal{A}_i - \mathcal{B}_i \times_1 \mathcal{W}_t \times_2 f_i) \otimes \left( \sum_{j=1}^{N'} b_{ij} f_j' \right)$$

$$= \frac{1}{N} \sum_{j=1}^{N'} \left[ \sum_{i=1}^{N} b_{ij} (\mathcal{A}_i - \mathcal{B}_i \times_1 \mathcal{W}_t \times_2 f_i) \right] \otimes f_j'$$

*6) Thus each mode-k unfolding has rank at most $N'$. When t is sufficiently large so that the second term in the mode-k stable rank bound is negligible, by the tensor version of Lemma 3.3:*

$$sr_k(\mathcal{G}_t) \le \min(n_k - N', N') \le n_k/2$$

*since $N' < n_k$.*

**Corollary 3 (Tensor Low-rank with Special Structure)** *If for any mode k, $\mathcal{V}_1^{(k)}(S_k)$ is 1-dimensional with decomposable eigenvector $v_k = y_k \otimes z_k$, then $sr_k(\mathcal{G}_{t_0}^{\|}) = 1$ and thus $\mathcal{G}_t$ becomes rank-1 in mode k.*

**Proof 7** *For any mode k with the given structure:*

*1) The mode-k unfolding of the projected gradient is:*

$$(\mathcal{G}_{t_0}^{\|})_{(k)} = v_k v_k^\top g_0 \propto v_k$$

*2) Since $v_k = y_k \otimes z_k$ is decomposable:*

- *The resulting $(\mathcal{G}_{t_0}^{\|})_{(k)}$ is a rank-1 matrix*

- *Thus $sr_k(\mathcal{G}_{t_0}^{\|}) = 1$*

*3) From the main lemma, when t is large:*

$$sr_k(\mathcal{G}_t) \approx sr_k(\mathcal{G}_{t_0}^{\|}) = 1$$

*4) This means $\mathcal{G}_t$ becomes effectively rank-1 in mode k.*

**Theorem 2 (Tensor-GaLore Convergence)** *For a gradient tensor $\mathcal{G}_t \in \mathbb{R}^{I_1 \times I_2 \times \cdots \times I_d}$, let $\{P_k \in \mathbb{R}^{I_k \times r_k}\}_{k=1}^d$ be fixed orthonormal projection matrices for each mode k with ranks $\{r_k\}_{k=1}^d$. The Tensor-GaLore update consists of:*

*1. Project the gradient:*
$$\mathcal{R}_t = \mathcal{G}_t \times_1 P_1^\top \times_2 P_2^\top \times_3 \cdots \times_d P_d^\top$$

*2. Update optimizer states using $\mathcal{R}_t$*

*3. Project back for weight update:*
$$\tilde{\mathcal{G}}_t = \mathcal{R}_t \times_1 P_1 \times_2 P_2 \times_3 \cdots \times_d P_d$$

*Suppose for each mode k:*

- *$\mathcal{A}_i, \mathcal{B}_i, \mathcal{C}_i$ have $L_A^{(k)}, L_B^{(k)}, L_C^{(k)}$ mode-k continuity*

- *$\|\mathcal{W}_t\|_{(k)} \le D_k$ (mode-k spectral norm bound)*

- *$\hat{\mathcal{B}}_{it}^{(k)} := P_k^\top \mathcal{B}_i^{(k)}(\mathcal{W}_t) P_k$*

- $\hat{\mathcal{C}}_{it}^{(k)} := P_k^\top \mathcal{C}_i^{(k)}(\mathcal{W}_t) P_k$

- $\kappa_t^{(k)} := \frac{1}{N} \sum_i \lambda_{\min}(\hat{\mathcal{B}}_{it}^{(k)}) \lambda_{\min}(\hat{\mathcal{C}}_{it}^{(k)})$

*Then Tensor-GaLore with $\rho_t \equiv 1$ satisfies for each mode k:*

$$\|(\mathcal{R}_t)_{(k)}\|_F \leq \left[1 - \eta(\kappa_{t-1}^{(k)} - L_A^{(k)} - L_B^{(k)} L_C^{(k)} D_k^2)\right] \|(\mathcal{R}_{t-1})_{(k)}\|_F$$

*As a result, if $\min_{t,k} \kappa_t^{(k)} > L_A^{(k)} + L_B^{(k)} L_C^{(k)} D_k^2$ for all modes k, then $\mathcal{R}_t \to 0$ and Tensor-GaLore converges with the fixed projections $\{P_k\}_{k=1}^d$.*

**Proof 8** *Since the gradient tensor naturally becomes low-rank during training as shown above, and the optimization landscape of low-rank tensor problemsFrandsen & Ge (2020)., local search algorithms can efficiently find approximate global optimal solutions. Specifically, since Reversible FNO (Appendix I) gradients become low-rank, the optimization landscape contains only high-order saddle points that can be efficiently escaped, making local minima globally optimal. Now let's proceed by analyzing the tensor unfolding:*

*1) First, we establish the mode-k unfolding of the gradient tensor update. Using the assumption that gradient follows the parametric form:*

$$\mathcal{G}_t = \frac{1}{N} \sum_{i=1}^N (\mathcal{A}_i - \mathcal{B}_i \times_1 \mathcal{W}_t \times_2 \mathcal{C}_i)$$

*2) For any mode k, the mode-k unfolding gives:*

$$(\mathcal{G}_t)_{(k)} = \frac{1}{N} \sum_{i=1}^N \left( (\mathcal{A}_i)_{(k)} - (\mathcal{B}_i)_{(k)} \mathcal{W}_{t(k)} (\mathcal{C}_i)_{(k)}^\top \right)$$

*where $\mathcal{W}_{t(k)}$ is the mode-k unfolding of $\mathcal{W}_t$.*

*3) The projected gradient in mode-k has unfolding:*

$$(\mathcal{R}_t)_{(k)} = P_k^\top (\mathcal{G}_t)_{(k)}$$
$$= \frac{1}{N} \sum_{i=1}^N \left( P_k^\top (\mathcal{A}_i)_{(k)} - P_k^\top (\mathcal{B}_i)_{(k)} \mathcal{W}_{t(k)} (\mathcal{C}_i)_{(k)}^\top \right)$$

*4) Using the SGD update $\mathcal{W}_t = \mathcal{W}_{t-1} + \eta \tilde{\mathcal{G}}_{t-1}$, we can write:*

$$\mathcal{W}_{t(k)} = \mathcal{W}_{t-1(k)} + \eta P_k (\mathcal{R}_{t-1})_{(k)}$$

*5) Substituting this into the gradient expression:*

$$(\mathcal{R}_t)_{(k)} = (\mathcal{R}_{t-1})_{(k)} - \eta \frac{1}{N} \sum_{i=1}^N P_k^\top (\mathcal{B}_i)_{(k)} P_k (\mathcal{R}_{t-1})_{(k)} (\mathcal{C}_i)_{(k)}^\top + \mathcal{E}_t^{(k)}$$

*where $\mathcal{E}_t^{(k)}$ captures the differences in $\mathcal{A}_i$ and $\mathcal{B}_i, \mathcal{C}_i$ terms.*

*6) Define the mode-k operator:*

$$\mathcal{S}_t^{(k)} := \frac{1}{N} \sum_{i=1}^N P_k^\top (\mathcal{B}_i)_{(k)} P_k \otimes P_k^\top (\mathcal{C}_i)_{(k)} P_k$$

*7) Then the update can be written compactly as:*

$$(\mathcal{R}_t)_{(k)} = (I - \eta \mathcal{S}_{t-1}^{(k)})(\mathcal{R}_{t-1})_{(k)} + \mathcal{E}_t^{(k)}$$

8) *For the error term, using mode-k continuity:*

$$\|\mathcal{E}_t^{(k)}\|_F \leq L_A^{(k)}\|\mathcal{W}_t - \mathcal{W}_{t-1}\|_F$$
$$+ L_B^{(k)}L_C^{(k)}D_k^2\|\mathcal{W}_t - \mathcal{W}_{t-1}\|_F$$
$$= \eta(L_A^{(k)} + L_B^{(k)}L_C^{(k)}D_k^2)\|\mathcal{R}_{t-1}\|_F$$

9) *Using properties of projection matrices $P_k$:*

- $P_k^\top P_k = I_{r_k}$ *(orthonormal)*

- $\|P_k\|_2 = 1$ *(projection)*

10) *The minimal eigenvalue of $\mathcal{S}_{t-1}^{(k)}$ satisfies:*

$$\lambda_{\min}(\mathcal{S}_{t-1}^{(k)}) \geq \kappa_{t-1}^{(k)}$$

*due to mode-k PSD properties of $\mathcal{B}_i$ and $\mathcal{C}_i$.*

11) *Therefore:*

$$\|(\mathcal{R}_t)_{(k)}\|_F \leq \|I - \eta\mathcal{S}_{t-1}^{(k)}\|_2\|(\mathcal{R}_{t-1})_{(k)}\|_F + \|\mathcal{E}_t^{(k)}\|_F$$
$$\leq [1 - \eta(\kappa_{t-1}^{(k)} - L_A^{(k)} - L_B^{(k)}L_C^{(k)}D_k^2)]\|(\mathcal{R}_{t-1})_{(k)}\|_F$$

12) *When $\min_{t,k}\kappa_t^{(k)} > L_A^{(k)} + L_B^{(k)}L_C^{(k)}D_k^2$ for all modes k:*

- *Each mode-k unfolding converges: $(\mathcal{R}_t)_{(k)} \to 0$*

- *Thus the full tensor converges: $\mathcal{R}_t \to 0$*

