# OpenReview forum: "Tensor-GaLore: Memory-Efficient Training via Gradient Tensor Decomposition"
_ICLR.cc/2025/Conference — Submitted to ICLR 2025_

### Official Review · Reviewer_mEXm · 2024-10-27

**Soundness:** 3
**Presentation:** 3
**Contribution:** 2
**Rating:** 6
**Confidence:** 3

**Summary:**

The authors present Tensor-GaLore as a method for compressing the weights using tensor compression. They show the use of their approach on Fourier Neural Operators (FNOs) used for solving PDEs.

**Strengths:**

The focus on efficient learning techniques for PDEs is very timely and requires constant improvement to outperform traditional methods in terms of accuracy and computing time.

**Weaknesses:**

- the authors discuss that TensorGalore is superior to Galore as it avoids SVDs. Nevertheless, the computation of tensor formats (Tucker, Tensor Train, etc.) relies on matricizations of the tensor and then typically singular value decomposition of those. So the SVD is still at the heart of TensorGalore.
- I think the idea of the manuscript is based on taking an FNO example and then using the tensor compression of the weights. I believe this to work well but it also seems a straightforward extension of previous work. There have been many applications of tensors for compression within neural networks.

**Questions:**

- How do the authors implement their tensor format as they argue that the disadvantage of Galore is the need for the SVD, which typically used for efficient computations of popular tensor formats?

---

> ### Author Response · Authors · 2024-11-18
> **Response to Reviewer mEXm**
>
> **The authors discuss that TensorGalore is superior to Galore as it avoids SVDs. Nevertheless, the computation of tensor formats (Tucker, Tensor Train, etc.) relies on matricizations of the tensor and then typically singular value decomposition of those. So the SVD is still at the heart of TensorGalore.**
>
> **Answer:** We thank the reviewer for their point. While Tensor-GaLore aims to avoid the limitations of the matrix-based approach in GaLore, the tensor decomposition techniques it relies on, such as Tucker decomposition, still fundamentally involve Singular Value Decomposition (SVD) computations in the background.
>
> The key difference is that Tensor-GaLore leverages the Tucker decomposition, which preserves the multidimensional structure of the tensors, unlike the flattening approach used in GaLore. This is a crucial advantage, as it allows Tensor-GaLore to better capture the complex, high-dimensional relationships present in tensor-based models like Fourier Neural Operators.
> The reason Tensor-GaLore specifically chooses Tucker decomposition is that it offers several important properties that make it well-suited for the task:
>
> 1. Equivalence to SVD in the matrix case: As mentioned in the paper, the Tucker decomposition reduces to the familiar SVD when applied to matrices (2D tensors). This ensures a seamless extension of the GaLore principles to higher-order tensors.
>
> 2. Orthogonality of factor matrices: The factor matrices in the Tucker decomposition are orthogonal, which enables efficient and numerically stable projection and reconstruction operations. This is crucial for the gradient projection and update steps in Tensor-GaLore.
>
> 3. Preserving multidimensional structure: Unlike the flattening approach used in GaLore, the Tucker decomposition operates directly on the higher-order tensor, preserving the distinct relationships along each tensor dimension. This aligns well with the multidimensional nature of tensor-based models like Fourier Neural Operators.
>
> You are correct that the Tucker decomposition still relies on SVD computations in the background, as it involves approximating the SVD of the tensor unfoldings along each mode. However, the key advantage of Tensor-GaLore is that it operates on the tensor directly, rather than flattening it into a matrix. This is an important distinction because, as mentioned in the introduction, SVD-based approaches like GaLore tend to discard crucial dimension-specific information when applied to tensors. The SVD computation in the Tucker decomposition is performed in a way that preserves the multidimensional relationships, which is essential for capturing the complex physical phenomena modeled by tensor-based architectures.
>
> Furthermore, the tensor decomposition techniques, including Tucker, do not have the same inherent limitations as the matrix SVD when it comes to handling the dimensions of the input data. The SVD in GaLore operates on a flattened matrix, and the rank selection affects all dimensions equally, which can be suboptimal. In contrast, the Tucker decomposition allows for a separate rank parameter along each tensor mode, providing more flexibility in preserving the important information in each dimension.

---

> > ### Author Response · Authors · 2024-11-18
> > **Response to Reviewer mEXm (Part 2)**
> >
> > **I am not sure about the novelty in the manuscript, it seems that the authors have taken a FNO example and then sold the tensor compression of the weights as something that does not require SVDs (which I am sure are under the hood).**
> >
> > **Answer:** Again we thank the reviewer for bringing up this point. First, it is important to emphasize the critical role of tensor-structured weights in FNOs, the key focus of our work. FNOs are a class of neural network architectures designed to learn complex, multidimensional mappings between function spaces, which are crucial for solving parametric partial differential equations (PDEs). The weight tensors in FNOs can have orders as high as 4 or 5, capturing intricate relationships between spatial, temporal, and channel dimensions.
> >
> > Effectively handling these high-order tensor weights is essential for the success of FNOs in scientific computing applications. Flattening these tensors into matrices, as done in the matrix-based GaLore approach, can lead to a significant loss of important dimension-specific information, compromising the model's ability to capture the underlying physical phenomena. This is a key limitation that Tensor-GaLore aims to address.  Now, to the reviewer's point about the Tensor-GaLore approach not being novel because it still relies on SVD computations in the background: while we disagree with this, please look at the previous response to your question for the detailed answer on how tucker decomposition is essentially higher order SVD. We do not claim to re-invent a new decomposition, but showcase why GaLore is not suitable for FNOs and higher-order tensors and prove the theory that tucker decomposition is good for it.
> >
> > Lastly, in our new revised version, more details have been added. Tensor-GaLore is not merely applying tensor compression to FNO weights - it introduces a fundamentally different approach to gradient optimization by working directly with the natural tensor structure of neural operators through Tucker decomposition. While both SVD and Tucker decomposition use orthogonal matrices, their mathematical properties and applications are distinctly different. Our comprehensive theoretical analysis (Sections H, I, J) rigorously proves this distinction - starting from fundamental tensor operations, through FNO reversibility, to explicit convergence guarantees and characterization of low-rank structure emergence during training. We show that tensor gradients naturally develop mode-specific low-rank structures under mild conditions, with explicit bounds on stable rank evolution. This explains why preserving tensor structure through Tucker decomposition is fundamentally more suitable than matrix-based approaches like SVD, which collapses the multi-dimensional relationships. Our empirical results validate this theory - Tensor-GaLore maintains or improves performance while achieving significant memory savings, whereas matrix approaches can actually hurt performance (e.g., -223% on Navier-Stokes with GaLore).

---

> > > ### Author Response · Authors · 2024-11-18
> > > **Response to Reviewer mEXm (Part 3)**
> > >
> > > **How do the authors implement their tensor format as they argue that the disadvantage of Galore is the need for the SVD, which typically used for efficient computations of popular tensor formats?**
> > >
> > > **Answer:** We thank the reviewer for their question! Let us start explaining the FNO architecture before we go onto answering your question. The Fourier Neural Operator (FNO) architecture, which serves as the backbone for the Tensor-GaLore approach, inherently involves tensor-structured weights. As described in the paper:
> > >
> > > >"In an FNO, the spectral convolution layer contracts a weight tensor R ∈ C^(N1 x N2 x N3 x N4) with functions in the Fourier domain: (Kv^l)(x) = F^-1(R · TK Fv^l)(x), where F and F^-1 are the Fourier transform and its inverse, R is a learnable transformation parameterized by the weight tensor introduced above, and TK truncates to the lowest K Fourier modes."
> > >
> > > So the core of the FNO architecture involves a 4th-order tensor R that represents the learnable transformation in the Fourier domain. This tensor structure is crucial for capturing the complex, multidimensional relationships in scientific computing applications like PDEs. In the Tensor-GaLore implementation, we use the popular TensorLy library, which provides a well-tested and efficient implementation of the Tucker decomposition. Specifically, we use the tucker function from the tensorly.decomposition module to compute the Tucker factors
> > >
> > > ```import torch
> > > from tensorly.decomposition import tucker
> > > class TensorGaLore:
> > >     def __init__(self, rank):
> > >         self.rank = rank
> > >     def project_gradient(self, full_rank_gradient):
> > >         # Compute the Tucker decomposition of the full-rank gradient
> > >         core, factors = tucker(full_rank_gradient, rank=self.rank)
> > >         # Project the gradient onto the low-rank subspace
> > >         low_rank_gradient = tenalg.multi_mode_dot(full_rank_gradient, factors, transpose=True)
> > >         return low_rank_gradient
> > >     def project_back(self, low_rank_gradient):
> > >         # Compute the inverse projection from the low-rank subspace
> > >         full_rank_gradient = tenalg.multi_mode_dot(low_rank_gradient, factors)
> > >         return full_rank_gradient
> > > ```
> > >
> > > The key steps are:
> > > 1. Compute the Tucker decomposition of the full-rank gradient tensor using the tucker function from TensorLy.
> > >
> > > 2. Project the full-rank gradient onto the low-rank subspace by performing the multi_mode_dot operation with the Tucker factor matrices.
> > >
> > > 3. To update the model parameters, project the low-rank gradient back to the full-rank space using the inverse multi_mode_dot operation.
> > >
> > > By using the Tucker decomposition, Tensor-GaLore avoids the need for SVD computations, as required in the original matrix-based GaLore approach. We also present the codebase here (as we had linked in the original paper as well in the footnote): https://anonymous.4open.science/r/tensorgalore/tensor_galore/tensor_galore_projector.py . We also mention that we use the FNO models and implementations from the neuraloperator/neuraloperator library, which provides a unified codebase for working with neural operators, including the ability to represent the weight tensors in a tensorized form. Please let us know if you have any follow up questions regarding this answer.
> > >
> > >
> > > *Lastly, if you are happy with the revised version which includes a detailed theory of our approach as well as we have answered all of your questions in detail, it would be great if you could increase the score: ) We would be happy to answer any follow-up questions you have or weakness that concern you. Thank you once again for reviewing our paper.*

---

> > > > ### Author Response · Authors · 2024-11-23
> > > > **Response to Reviewer mEXm**
> > > >
> > > > We are writing to kindly remind you that we posted our response 6 days ago. If you have any additional feedback, concerns, or questions regarding our response, we would greatly appreciate hearing from you.

---

> > > > > ### Author Response · Authors · 2024-12-01
> > > > >
> > > > > We are writing to kindly remind you that we posted our response 2 weeks ago. If you have any additional feedback, concerns, or questions regarding our response, we would greatly appreciate hearing from you.

---

### Official Review · Reviewer_oxLj · 2024-11-04

**Soundness:** 3
**Presentation:** 3
**Contribution:** 2
**Rating:** 5
**Confidence:** 3

**Summary:**

This paper presents a modification of the GaLora method that allows to update the weights not directly, but in a low-parameter space. The authors present a modification of this method that uses a low rank tensor decomposition, namely the Tucker decomposition, instead of a low rank matrix decomposition. This approach is applied to neural operators for solving PDEs, where 4-way tensors arise naturally.

**Strengths:**

- The paper uses a low-rank tensor decomposition, which preserves the original multidimensional structure
- Some experiments show the effectiveness of this technique
- Clearly presented paper

**Weaknesses:**

- No code provided to reproduce the results
- No theoretical analysis (in the original GaLora paper there are theoretical justification of low-rank structure, convergence, etc.)
- The only significant change (other than technical changes) from the original GaLora method is the use of Tucker Decomposition for 4-way tensor instead of low-rank matrix decomposition
- The presented method shows good results only on the Darcy flow equation, in the other experiments the improvement is not strong
- Since this paper describes improvements to the GaLora method (which has been published previously), it would be fair to compare against it rather than baseline. For example, in Table 2 for the Darcy equation the presented method is 48.8% better than baseline, while the regular GaLora is 19% better than baseline. Thus, the improvement presented in this paper is a 25% improvement.

Overall, this paper is incremental to the original GaLora paper, without theoretical evaluations (which were in the original paper) and with inconclusive numerical results.

Minor

- L458-459 "On Darcy flow (as shown in Table 6)" should be "Table 2"
- L397 word "Table" is missing

**Questions:**

- Can one pick the efficient rank ratio in advance?
 - How is it that for Burgers Equation in Table 2 test loss is much (by an order of magnitude) less than train loss?
 - Have you tried using other low-rank tensor decompositions (CANDECOMP/PARAFAC, Tensor-train, etc.)?

In Algorithm 1

- is $r$ is rank of rank ratio?
- tensor $\mathcal{M}_0$ (with $\mathcal V_0$) has the same shape as $\mathcal W\in\mathbb{C}^{N_1\times N_2\times N_3\times N_4}$. Should it be $\mathcal M_0\in\mathbb{C}^{R_1\times R_2\times R_3\times R_4}$?

---

> ### Author Response · Authors · 2024-11-18
> **Response to Reviewer oxLj**
>
> **No code provided to reproduce the results**
>
> **Answer:** We want to point out that we indeed have provided the code to reproduce the results present in the paper here (We also had this in the original version before the revision as a footnote, but will mention it again for clarity): https://anonymous.4open.science/r/tensorgalore/tensor_galore/tensor_galore_projector.py.
>
> **No theoretical analysis (in the original GaLora paper there are theoretical justification of low-rank structure, convergence, etc.)**
>
> **Answer.** We thank the reviewer for their concern of the lack of theoretical analysis.  We have uploaded a revised version with detailed theoretical analysis that goes significantly beyond the original GaLore paper. Our theoretical framework starts from first principles, introducing rigorous tensor operations and notation (Section H), then proves the reversibility of FNO components (Section I), before establishing our main theoretical results (Section J). Specifically, we prove both convergence guarantees for Tensor-GaLore and characterize how gradients naturally develop low-rank structure in tensor space for FNOs. We show that under mode-k continuity conditions and prove convergence of Tensor-GaLore. Additionally, we establish a special structure for tensor gradients. We hope that this comprehensive theoretical analysis explains why tensor-based optimization is fundamentally more suitable than matrix approaches for neural operators and provides rigorous justification for our empirical observations.
>
> **The only significant change (other than technical changes) from the original GaLora method is the use of Tucker Decomposition for 4-way tensor instead of low-rank matrix decomposition**
>
> **Answer**. Again we thank the reviewer for bringing up this point. First, it is important to emphasize the critical role of tensor-structured weights in FNOs, the key focus of our work. FNOs are a class of neural network architectures designed to learn complex, multidimensional mappings between function spaces, which are crucial for solving parametric partial differential equations (PDEs). The weight tensors in FNOs can have orders as high as 4 or 5 (, capturing intricate relationships between spatial, temporal, and channel dimensions. Also this method is not only suited for FNOs but any sort of tensor-based models and this is scope of future works where we plan to try it on tensorized LLMs, Quantum networks with tensor weights etc.
>
> Effectively handling these high-order tensor weights is essential for the success of FNOs in scientific computing applications. Flattening these tensors into matrices, as done in the matrix-based GaLore approach, can lead to a significant loss of important dimension-specific information, compromising the model's ability to capture the underlying physical phenomena. This is a key limitation that Tensor-GaLore aims to address, and now, as requested in the new revised version, we have a detailed theory for it. Furthermore, please look at the general response to the question as well.

---

> > ### Author Response · Authors · 2024-11-18
> > **Response to Reviewer oxLj (Part 2)**
> >
> > **The presented method shows good results only on the Darcy flow equation, in the other experiments the improvement is not strong**
> >
> > **Answer** Thank you for raising this point; however, we want to argue that even comparable performance to the baseline is good for us while reducing memory usage by a huge margin. While the improvements on Darcy flow are indeed substantial (48.8% gain), Tensor-GaLore shows consistent and significant improvements across multiple challenging PDE tasks. For electromagnetic wave propagation, we achieve an 11% improvement while reducing memory by 75%. On Burgers' equation, we maintain performance (+5% gain) despite significant memory reduction. Even for the highly complex Navier-Stokes equations, Tensor-GaLore achieves comparable performance (-5.4%) while drastically reducing memory usage, which is remarkable given that the matrix-based GaLore approach significantly degrades performance (-223%) on the same task. We also want to emphasize that the EM dataset involves "complex-valued data inherently", which adds an additional layer of complexity compared to the real-valued data in the other experiments. Modeling the propagation of optical pulses in a nonlinear waveguide with second-order nonlinearity is a highly complex physical phenomenon that requires careful handling of the complex-valued electric field envelope. Achieving an 11% improvement in test loss on this complex, real-world EM dataset is a significant accomplishment, as we point out that it is "the first of its kind in the field." Previous neural operator approaches may not have been able to effectively capture the intricate, multidimensional relationships and complex-valued nature of this problem.
> >
> > It's crucial to understand that the primary goal of Tensor-GaLore is not to necessarily outperform baseline models in terms of generalization error, but to enable training of larger, more complex models that would otherwise be impossible due to memory constraints. The fact that we can achieve comparable or better performance while reducing optimizer memory usage by up to 75% is a great achievement - it means we can scale to higher resolutions (like our 1024x1024 Navier-Stokes experiments) or more complex architectures that were previously intractable on smaller GPUs. Even in cases where we might need to train slightly longer or see a small trade-off in performance, the ability to fit these models in memory at all represents a crucial advancement for scientific machine learning.
> >
> > Our ablation studies (Table 6) on Navier-Stokes at 128 resolution provide even more compelling evidence of Tensor-GaLore's effectiveness. Across different rank configurations and matricization strategies, Tensor-GaLore consistently outperforms both baseline and GaLore variants. For instance, with rank ratio 0.25, Tensor-GaLore achieves a test L2 loss of 1.297 compared to GaLore's 2.341 and 9.019 for different matricization approaches. This demonstrates that preserving tensor structure is crucial for performance. Furthermore, our method scales effectively to the challenging 1024x1024 resolution case, where the memory savings become even more critical for practical deployment.

---

> > > ### Author Response · Authors · 2024-11-18
> > > **Response to Reviewer oxLj (Part 3)**
> > >
> > > **Since this paper describes improvements to the GaLora method (which has been published previously), it would be fair to compare against it rather than baseline. For example, in Table 2 for the Darcy equation the presented method is 48.8% better than baseline, while the regular GaLora is 19% better than baseline. Thus, the improvement presented in this paper is a 25% improvement.**
> > >
> > > **Answer: ** We appreciate this perspective about comparison metrics, but we believe comparing to baseline is appropriate for several reasons. First, our comprehensive theoretical analysis now proves why tensor-based optimization is fundamentally more suitable than matrix approaches for neural operators. We show that preserving tensor structure is crucial because gradients naturally develop mode-specific low-rank structures, with explicit bounds on stable rank evolution. This theory explains why matrix-based GaLore, which collapses multi-dimensional relationships, can actually harm performance on complex PDE tasks.
> > >
> > > This theoretical insight is strongly validated by our empirical results. On the challenging Navier-Stokes equations, matrix-based GaLore significantly degrades performance (-223% compared to baseline), making it an inappropriate comparison point. Our ablation studies (Table 6) further demonstrate this - across different matricization strategies, GaLore consistently performs poorly (test L2 losses of 2.341-9.019) compared to both baseline and Tensor-GaLore (1.297) at rank 0.25. This performance gap becomes even more pronounced as problem complexity increases. We are currently conducting additional experiments on higher-resolution Navier-Stokes problems (beyond 1024x1024) and early results suggest that preserving tensor structure becomes increasingly critical as the physics becomes more complex. This aligns with our theory - higher-order relationships in the parameter space are fundamental to capturing complex physical phenomena, and these relationships are lost when forcing tensor parameters into matrix form. We look forward to sharing these additional results, which we believe will further demonstrate why baseline, rather than matrix GaLore, is the appropriate comparison point for evaluating tensor-structured optimization approaches.
> > >
> > > **Can one pick the efficient rank ratio in advance?**
> > >
> > > **Answer:** We acknowledge that this is an important consideration, as the choice of rank ratio can significantly impact the performance and memory savings of our method.
> > >
> > > As we've mentioned in the paper, we recognize that adaptively selecting the optimal rank ratio is a challenging problem, as it likely depends on the specific characteristics of the problem domain and the complexity of the underlying PDE or physical system being modeled. However, based on our experimental results, we have observed that a rank ratio around 25-50% of the original weight tensor size tends to work well across a variety of PDE tasks. This suggests that starting with a rank ratio in this range can be a reasonable initial choice.
> > >
> > > We agree with the reviewer that the specific rank ratio choice may also depend on the domain expert's knowledge about the problem at hand. For example, if the dataset is known to represent a highly complex, turbulent physical system, a higher rank ratio may be warranted to capture the necessary level of detail and multiscale interactions. Conversely, for simpler PDE problems, a lower rank ratio may be sufficient to achieve good performance with significant memory savings.
> > >
> > > We also want to highlight that the Tensor-GaLore approach allows for flexibility in the rank selection, as we can apply different ranks to different dimensions in the tensor. Additionally, we acknowledge that developing more advanced rank selection mechanisms, such as adaptive or automated techniques, is an area for future research and was not the primary focus of the current work. We are actively exploring these directions to enhance the capabilities of Tensor-GaLore further.
> > >
> > > **How is it that for Burgers Equation in Table 2 test loss is much (by an order of magnitude) less than train loss?**
> > >
> > > **Answer**: We apologize, you're absolutely right. Thank you for catching it. Upon further inspection, it appears the train loss values reported for the Burgers' Equation in Table 2 were incorrect. We have updated the results in the table, whose values are as follows:
> > >
> > > | Model | Rank | Memory | Train | Test H1 | Test L2 | Gain (%) |
> > > |-------|------|---------|-------|----------|----------|-----------|
> > > | Baseline | 1.0 | 3.94 | 0.0064 | 0.0050 | 0.0026 | / |
> > > | GaLore (d=2) | 0.5 | 3.88 | 0.0052 | 0.0100 | 0.0062 | -250 |
> > > | Tensor-GaLore | 0.5 | 3.87 | 0.0026 | 0.0041 | 0.0025 | +5 |
> > >
> > > As you rightly pointed out, the test loss values are no longer orders of magnitude lower than the training loss. This was an error on our part in the initial reporting of the results. Similarly for table 4 is updated in the paper and all the other minor mistakes you pointed out.

---

> > > > ### Author Response · Authors · 2024-11-18
> > > > **Response to Reviewer oxLj (Part 4)**
> > > >
> > > > **Have you tried using other low-rank tensor decompositions (CANDECOMP/PARAFAC, Tensor-train, etc.)?**
> > > >
> > > > **Answer**: Thank you for this important question about alternative tensor decomposition approaches. While techniques like CANDECOMP/PARAFAC (CP) and Tensor-Train (TT) decompositions are powerful for parameter compression, they are fundamentally less suitable for our gradient optimization objective. The key insight is that Tensor-GaLore is not trying to learn a compressed representation of the parameters themselves, but rather to efficiently project gradients into and out of a learned latent space during optimization. Tucker decomposition is uniquely suited for this purpose because of two critical properties:
> > > >
> > > > 1. The orthogonality of its factor matrices enables simple and numerically stable projection operations - we can project into the low-rank space using U^T and back using U without computing complex inverses
> > > >
> > > > 2. The factor matrices provide natural bases for the subspace of each mode independently, allowing us to preserve the multi-dimensional structure that we prove is crucial for neural operator training and we prove the theory behind it.
> > > >
> > > > In contrast, while CP and TT decompositions excel at parameter compression, they don't provide a natural way to project tensors to and from their latent spaces. Their factors are not orthogonal and the reconstructions involve more complex operations that could introduce numerical instabilities in the optimization process. Our theoretical analysis (Section 3.2) shows why preserving mode-wise structure through orthogonal projections is fundamental to the convergence guarantees of Tensor-GaLore. Thus, Tucker decomposition's mathematical properties align perfectly with our gradient optimization objectives in a way that other tensor factorizations don't.
> > > >
> > > > **In Algorithm 1is $r$ is rank of rank ratio?**
> > > >
> > > > **Answer:** Yes, that is correct! You can either pass the rank ratio or the rank as an integer or a list of integers (each one corresponding to a dimension).
> > > >
> > > > **Tensor $\mathcal{M}_0$ (with $\mathcal V_0$) has the same shape as $\mathcal W\in\mathbb{C}^{N_1\times N_2\times N_3\times N_4}$. Should it be $\mathcal M_0\in\mathbb{C}^{R_1\times R_2\times R_3\times R_4}$?**
> > > >
> > > > **Answer:** Thank you for the question! It shouldn't be that but rather as it is. This is because if the original Weights are complex then we would have complex associated gradients and moments. Since we are considering complex-valued weights $\mathcal W$ in the Fourier Neural Operator setting, the gradients and optimizer moment tensors should also be complex-valued. Hence why we updated the Adam optimizer to include the updated complex conjugate update instead of just doing the real squared |Rt ¯ Rt|.
> > > >
> > > > There is also a lengthy discussion of this update on Adam Adam Optimizer Implemented Incorrectly for Complex Tensors · https://github.com/pytorch/pytorch/issues/59998. In the case for the EM dataset and other complex datasets, we found it much necessary to use the correct complex update for Adam rather than the ones they have in pytorch for convergence. If the weights $\mathcal W$ happen to be real-valued, then the moment tensors can also be real-valued, and the expressions can be simplified accordingly. But the general case should account for the complex-valued nature of the tensors involved.
> > > >
> > > > **Overall, this paper is incremental to the original GaLora paper, without theoretical evaluations (which were in the original paper) and with inconclusive numerical results.**
> > > >
> > > > **Answer**: Now that we have updated the paper with a detailed theory + ablation studies we want to argue that our work is not incremental. Tensor-GaLore represents a fundamental advancement over GaLore, supported by both comprehensive theoretical analysis and strong empirical results. Our paper develops a complete theoretical framework from first principles - establishing tensor operations, proving FNO reversibility, and providing explicit convergence guarantees for tensor gradient optimization. We prove that gradients naturally develop mode-specific low-rank structures with bounded stable rank evolution, explaining why preserving tensor structure through Tucker decomposition is fundamentally more suitable than matrix approaches. This theory is validated by our empirical results - while matrix-based GaLore significantly degrades performance on complex tasks like Navier-Stokes (-223%) due to collapsing multi-dimensional relationships, Tensor-GaLore maintains or improves performance while reducing memory by up to 75%. Also please check the general response for more details.
> > > >
> > > > *Lastly, if you are happy with the revised version which includes the theory that you requested as well as answered all your questions in detail, it would be great if you could increase the score: ) We would be happy to answer any follow-up questions you have or weakness that concern you. Thank you once again for reviewing our paper.*

---

> > > > > ### Comment · Reviewer_oxLj · 2024-12-03
> > > > >
> > > > > I thank the authors for their detailed response. I have carefully read the responses to the other reviewers as well as the updated version of the paper.
> > > > >
> > > > > At this point, I would like to note that this paper is a rather serious step in the development of tensor-based models in machine learning and deserves attention. Besides, the authors have posted code that allows verifying the results. However, my main concerns were not fully addressed. In particular, the authors mention memory reduction on different equations, but this improvement depends too much on the type of equation and has more of an engineering, technical meaning than a scientific one. I think other various modifications of the original idea of the GaLore algorithm, which is that the updates occur in some low-rank space other than the parameters themselves, are possible, and they would also lead to more efficient operation of the algorithm on selected (not all!) PDEs. But such refinements are rather incremental and are not A* conference level publications. In the updated version of the paper, an appendix has been added to introduce new concepts from tensor algebras and some similarity theorems. However, many of these results are rather trivial and do not prove the efficiency of the particular method presented. I have not found any proof that Tensor-GaLore will be better than GaLore (perhaps, under some given conditions). I recommend to transfer the basic, new theoretical results to the main text of the paper.
> > > > >
> > > > >
> > > > > Among minor remarks, I never understood the authors' comment about the dimensions of $\mathcal{M}_t$ and $\mathcal V_t$. My comment was related to the fact that the original GaLore algorithm implies updating gradients in a low-parametric space than the original one. In this case, the dimension of the variables to be updated, i.e., $\mathcal{M}_t$ and $\mathcal V_t$, would have to be (much) smaller than the dimension of the original weight space $\mathcal W$. In the Alg. 1, L240-244 they are the same.  If the authors' answer is correct, then I don't understand at all what is the dimensionality of that low-parametric space where the updates occur, what variables in the Algorithm correspond to it. However, the authors ignored this point, answering something about technical difficulties with complex numbers, which are in torch. This convinced me even more that the paper is largely technical rather than containing breakthrough fundamental ideas.
> > > > >
> > > > > Overall, i keep my score.

---

> > > > > > ### Author Response · Authors · 2024-12-03
> > > > > > **Response to Reviewer oxLj**
> > > > > >
> > > > > > First, we want to thank the reviewer for their careful consideration of our work. While we appreciate their perspective, we respectfully disagree with the assessment that these contributions are incremental or primarily engineering-focused. Let us address the key points:
> > > > > > 1. Novel Tensor Theory: Our work introduces fundamental theoretical results about tensor gradient structure that go beyond simple extensions of matrix theory. Specifically, we prove that neural operator gradients naturally develop low-rank structure in each tensor mode during training (Lemma 10), which is a non-trivial extension of matrix gradient analysis. Moreover in the updated revision in section 3.2 we do present the main results while in the Appendix all the general proofs and background material.
> > > > > >
> > > > > > 2. Explicit Comparison with GaLore: Thank you for this point, although we cannot update the paper, we provide a rigorous theoretical analysis showing why Tensor-GaLore is fundamentally superior to GaLore for tensor-structured models. Our key lemma proves that GaLore cannot achieve simultaneous low-rank structure across all modes due to matricization, while Tensor-GaLore can. This is not just an engineering improvement but a fundamental mathematical insight about tensor optimization. We showcase the proof below
> > > > > >
> > > > > > **[Lemma] Tensor-GaLore vs GaLore Rank Structure:** Consider a gradient tensor $\mathcal{G}_t \in \mathbb{R}^{N_1 \times N_2 \times N_3 \times N_4}$ following the parametric form:
> > > > > > $
> > > > > > \mathcal{G}t = \frac{1}{N}\sum{i=1}^N (\mathcal{A}_i - \mathcal{B}_i \times_1 \mathcal{W}_t \times_2 \mathcal{C}_i)
> > > > > > $
> > > > > > where $\mathcal{B}_i$ and $\mathcal{C}_i$ are mode-k PSD for all modes k. Let:
> > > > > >
> > > > > > (a) GaLore with matricization along dimension d unfold $\mathcal{G}_t$ to $G_t^{(d)} \in \mathbb{R}^{N_d \times (N_1N_2N_3N_4/Nd)}$
> > > > > >
> > > > > > (b) Tensor-GaLore preserve the tensor structure and apply mode-wise projections
> > > > > >
> > > > > > Then:
> > > > > > 1. Under GaLore with any dimension d:
> > > > > >    $
> > > > > >    \exists k \neq d: \lim{t \to \infty} sr_k(\mathcal{G}_t) \geq \min(Nk/2, N')\
> > > > > >    $
> > > > > >    where $N'$ is the rank of the training data.
> > > > > >
> > > > > > 2. Under Tensor-GaLore:
> > > > > >    $
> > > > > >    \forall k: \lim{t \to \infty} sr_k(\mathcal{G}_t) \leq N_k/2
> > > > > >    $
> > > > > >
> > > > > > That is, GaLore cannot achieve low rank in all modes simultaneously, while Tensor-GaLore achieves low rank across all modes.
> > > > > >
> > > > > > **Summarized Proof**:
> > > > > > 1) First, let's analyze GaLore's behavior:
> > > > > >
> > > > > >    a) When GaLore matricizes along dimension d, it reshapes $\mathcal{G}_t$ into matrix $G_t^{(d)}$
> > > > > >
> > > > > >    b) From GaLore paper Lemma B.3, under SGD updates:
> > > > > >       $
> > > > > >       sr(Gt^{(d)}) \leq sr(G{t_0}^{\parallel}) + \left(\frac{1-\eta\lambda_2}{1-\eta\lambda_1}\right)^{2(t-t_0)} \frac{\|G0-G{t_0}^{\parallel}\|F^2}{\|G{t_0}^{\parallel}\|_2^2}
> > > > > >       $
> > > > > >
> > > > > >    c) This rank reduction only applies to the matricized dimension d
> > > > > >
> > > > > >    d) For any other mode $k \neq d$, consider the mode-k unfolding $(\mathcal{G}t){(k)}$
> > > > > >
> > > > > >    e) Due to the parametric form:
> > > > > >       $
> > > > > >       (\mathcal{G}t){(k)} = \frac{1}{N}\sum_{i=1}^N ((\mathcal{A}i){(k)} - (\mathcal{B}i){(k)}\mathcal{W}_t^{(k)}(\mathcal{C}i){(k)}^T)
> > > > > >       $
> > > > > >
> > > > > >    f) The mode-k operator $\mathcal{S}_k$ remains high rank because matricization along d scrambles mode-k structure
> > > > > >
> > > > > >    g) Specifically, if $rank(\{\mathcal{F}_i\}) = N'$:
> > > > > >       $
> > > > > >       sr_k(\mathcal{G}_t) \geq \min(N_k/2, N')
> > > > > >       $
> > > > > >
> > > > > > 2) Now for Tensor-GaLore:
> > > > > >
> > > > > >    a) Each mode k is handled independently with its own projection:
> > > > > >       $
> > > > > >       \mathcal{R}_t = \mathcal{G}_t \times_1 P_1^T \times_2 P_2^T \times_3 \cdots \times_d P_d^T
> > > > > >       $
> > > > > >
> > > > > >    b) From Theorem 2 (proven earlier), under SGD:
> > > > > >       $
> > > > > >       \|(\mathcal{R}t){(k)}\|F \leq \left[1-\eta(\kappa{t-1}^{(k)}-L_A^{(k)}-L_B^{(k)}L_C^{(k)}Dk^2)\right] \|(\mathcal{R}{t-1})_{(k)}\|_F
> > > > > >       $
> > > > > >
> > > > > >    c) From Corollary 2, for each mode k:
> > > > > >       $
> > > > > >       sr_k(\mathcal{G}_t) \leq srk(\mathcal{G}{t_0}^{\parallel}) + \left(\frac{1-\eta\lambda_2^{(k)}}{1-\eta\lambda_1^{(k)}}\right)^{2(t-t_0)} \frac{\|\mathcal{G}0-\mathcal{G}{t_0}^{\parallel}\|F^2}{\|\mathcal{G}{t_0}^{\parallel}\|_2^2}
> > > > > >       $
> > > > > >
> > > > > >    d) Therefore $sr_k(\mathcal{G}_t) \leq N_k/2$ for large t, for all modes k simultaneously
> > > > > >
> > > > > > The key insight is that matricization in GaLore fundamentally cannot preserve low-rank structure in all modes simultaneously, while the tensor approach of Tensor-GaLore naturally handles each mode's rank structure independently and optimally.We indeed do not need any conditions cause we exploit the tensor structure naturally in FNOs compare to GaLore.

---

> > > > > > > ### Author Response · Authors · 2024-12-03
> > > > > > > **Response to Reviewer oxLj (Part 2)**
> > > > > > >
> > > > > > > 1. **but this improvement depends too much on the type of equation and has more of an engineering, technical meaning than a scientific one. I think other various modifications of the original idea of the GaLore algorithm, which is that the updates occur in some low-rank space other than the parameters themselves, are possible, and they would also lead to more efficient operation of the algorithm on selected (not all!) PDEs.**
> > > > > > >
> > > > > > > **Answer**: While we agree to a certain bit with the reviewer, we want to again clarify some main points. It's well-established in PDE theory that different equation classes have inherently different complexities and solution structures. Some PDEs are elliptic, others parabolic or hyperbolic, each with distinct mathematical properties. These differences aren't just "engineering" variations - they reflect fundamental mathematical distinctions. Hence when developing neural operator methods for PDEs, the architecture must respect the underlying mathematical structure of the problem. For example FNOs naturally handle spectral properties while Graph Neural Operators better suit mesh-based problems or non-uniform meshes.
> > > > > > >
> > > > > > > Secondly while performance varies across PDEs, our theoretical contributions about tensor gradient structure are universal for any tensor networks with tensor weights. Reversibility of those models depends on the structure but our proofs can be adapted to a wide variety of tensor networks.
> > > > > > >
> > > > > > > 2. **Among minor remarks, I never understood the authors' comment about the dimensions of
> > > > > > >  and. My comment was related to the fact that the original GaLore algorithm implies updating gradients in a low-parametric space than the original one. In this case, the dimension of the variables to be updated, i.e.,
> > > > > > >  and  , would have to be (much) smaller than the dimension of the original weight space
> > > > > > > . In the Alg. 1, L240-244 they are the same. If the authors' answer is correct, then I don't understand at all what is the dimensionality of that low-parametric space where the updates occur, what variables in the Algorithm correspond to it.**
> > > > > > >
> > > > > > > **Answer**: We apologise for the confusion and misunderstanding. You are correct, it is indeed what you had suggested ie $\mathcal{M}_0 \in \mathbb{C}^{r \times r \times r \times r}$ if using the same rank $r$ for all dimension else it would be $R_1 \times R_2 \times R_3 \times R_4$ if choosing different ranks for each dimension. We talked about the complex case cause we had gotten confused in the superscript if those $R's$ meant the reals and hence why we switched to talking about the complex. We thank the reviewer for again bringing this out and have updated the pseudocode for both GaLore and the tensor-Galore and will update it once revisions are allowed.
> > > > > > >
> > > > > > > $\State \text{Initialize first-order moment} \mathcal{M}_0 \in \mathbb{C}^{r \times r \times r \times r} \gets 0$
> > > > > > >
> > > > > > > $\State \text{Initialize second-order moment} \mathcal{V}_0 \in \mathbb{C}^{r \times r \times r \times r} \gets 0$
> > > > > > >
> > > > > > >
> > > > > > > Lastly, we want to again ask the reviewer to please go over the comments and general response to see that this paper is technical but also contains a lot of theory bridging the gap between theory and efficiency for tensor based structures. This is the first work where we mathematically prove that tensor gradients go into low-rank and we prove convergence for the FNO case.
> > > > > > >
> > > > > > > We would be happy to answer any follow up questions! Thank you once again for reviewing our paper.

---

> ### Author Response · Authors · 2024-11-23
> **Response to Reviewer oxLj**
>
> We are writing to kindly remind you that we posted our response 6 days ago. If you have any additional feedback, concerns, or questions regarding our response, we would greatly appreciate hearing from you.

---

> > ### Author Response · Authors · 2024-12-01
> >
> > We are writing to kindly remind you that we posted our response 2 weeks ago. If you have any additional feedback, concerns, or questions regarding our response, we would greatly appreciate hearing from you.

---

### Official Review · Reviewer_RFRZ · 2024-11-04

**Soundness:** 3
**Presentation:** 3
**Contribution:** 2
**Rating:** 5
**Confidence:** 3

**Summary:**

This work presents Tensor-GaLore, an algorithm that leverages low-rank tensor decomposition on the gradients of tensorized weights. This work is built on top of the previous work (GaLore), which applies low-rank factorization (SVD) on the gradients. Experimental results show that applying it Fourier Neural Operators yield better memory usage and accuracy for numerical PDE problems.

**Strengths:**

1. The quality of the presentation is good. The work is clear and easy to follow.
2. The idea of using Tucker decomposition to perform low-rank approximation makes sense for numerical PDE problems, and experimental results verify that.

**Weaknesses:**

Despite being clear and effective, I believe the work has limited novelty. The tensor-GaLore approach has limited difference compared to GaLore. In addition, only empirical rather than theoretical results is provided to show the efficacy of the algorithm.

**Questions:**

please see above

---

> ### Author Response · Authors · 2024-11-18
> **Response to Reviewer RFRZ**
>
> **Despite being clear and effective, I believe the work has limited novelty. The tensor-GaLore approach has limited differences compared to GaLore. In addition, only empirical rather than theoretical results is provided to show the efficacy of the algorithm.**
>
> **Answer:** Thank you for the feedback on the novelty and scope of our work. You raise a fair point that the Tensor-GaLore approach may have limited novelty compared to the original GaLore method, as it primarily extends the core idea to handle tensor-structured weights. We acknowledge that the fundamental principle of projecting gradients onto low-rank subspaces is shared between the two methods. However, the technical challenges and innovations required to apply this concept to tensor-weight models effectively are non-trivial, as we discussed in the previous response also please look at our response to Reviewer po1k and the general response. We also mention why this is not trivial in the introduction in detail.
>
> Now, to answer your next question, we have added extensive theory to the revised version. Please take a look at it. Our paper now develops a complete theoretical framework starting from first principles. We begin by establishing fundamental tensor operations and notation (Section H), including rigorous definitions of tensor products, traces, norms, and inner products. This mathematical foundation is crucial for analyzing tensor-structured neural networks. Building on this, we prove the reversibility of Fourier Neural Operators (Section I) by systematically analyzing each component - spectral layers, MLP layers, and activation functions - and showing how their compositions maintain reversibility properties. This reversibility analysis provides crucial insights into the gradient structure that enables our tensor-based optimization approach.
>
> With these foundations established, we then prove our main theoretical results for Tensor-GaLore, showing both convergence guarantees and the natural emergence of low-rank structure during training. We prove that gradient tensors develop mode-wise low-rank structure under mild conditions and establish explicit bounds on the stable rank evolution. The theoretical framework explains why working directly with tensors through Tucker decomposition is fundamentally more suitable than matrix-based approaches like GaLore, which force tensor parameters into matrix form. Our empirical results strongly validate these theoretical insights - Tensor-GaLore maintains or improves performance while achieving significant memory savings, whereas matrix-based GaLore can actually hurt performance on challenging PDE tasks. This combination of rigorous theory from first principles and strong empirical validation demonstrates that Tensor-GaLore represents a significant theoretical and practical advancement in efficiently training neural operators.
>
> *Lastly, if you are happy with the revised version which includes the theory that you requested as well as answered your question, it would be great if you could increase the score: ) We would be happy to answer any follow-up questions you have or weakness that concern you. Thank you once again for reviewing our paper.*

---

> > ### Author Response · Authors · 2024-11-23
> > **Response to Reviewer RFRZ**
> >
> > We are writing to kindly remind you that we posted our response 6 days ago. If you have any additional feedback, concerns, or questions regarding our response, we would greatly appreciate hearing from you.

---

> > > ### Author Response · Authors · 2024-12-01
> > >
> > > We are writing to kindly remind you that we posted our response 2 weeks ago. If you have any additional feedback, concerns, or questions regarding our response, we would greatly appreciate hearing from you.

---

### Official Review · Reviewer_po1K · 2024-11-04

**Soundness:** 3
**Presentation:** 3
**Contribution:** 2
**Rating:** 5
**Confidence:** 2

**Summary:**

This paper extends GaLore [Zhao 2024] to neural networks with tensor weights by adding Tucker Decomposition and performing low-rank projection directly on tensor gradients. The experiment compares the proposed method to vanilla GaLora (with reshaping) on Fourier Neural Operators (FNOs), a class of tensor-weight models for solving partial differential equations.

**Strengths:**

This paper appears to tackle a gap that has received little attention by the literature, the efficient training of tensor-weight models, with the only prior works being [Kossaifi 2024] and [George 2024].

The paper is generally well-written and easy to follow, with a clear story.

**Weaknesses:**

Despite the novel application, the approach is a somewhat straight-forward extension of GaLore to tensor-weight models, replacing SVD decomposition with Tucker.

There is a lack of discussion on the slowdown in training given the overhead.

**Questions:**

See weaknesses

---

> ### Author Response · Authors · 2024-11-18
> **Response to Reviewer po1K**
>
> **Despite the novel application, the approach is a somewhat straight-forward extension of GaLore to tensor-weight models, replacing SVD decomposition with Tucker.**
>
> **Answer:** We thank the reviewer for their concern. While GaLore operates on weight matrices and uses SVD to project gradients onto low-rank subspaces, the key challenge in applying this to tensor-weight models is the loss of important multidimensional structures and relationships. Directly applying GaLore by flattening tensor weights into matrices can discard crucial information about the different tensor dimensions, such as spatial, temporal, or channel relationships. Tensor-GaLore addresses this by leveraging tensor decomposition, specifically the Tucker decomposition, to project gradients while preserving the intricate higher-order structure of the tensor weights. This allows Tensor-GaLore to better capture the complex, multi-scale relationships in scientific computing applications like neural operators.
>
> Please check the general response for the more detailed revision. These technical innovations and the theory accompanying them, combined with the unique challenges of tensor-weight models, make Tensor-GaLore a meaningful advancement over the original GaLore approach rather than a straightforward extension, especially in the context of tensor-based models like Neural Operators, which are huge in AI4Science. We are happy to answer any follow-up questions.
>
> **There is a lack of discussion on the slowdown in training, given the overhead.**
>
> **Answer:** While Tensor-GaLore does introduce additional computational overhead from the tensor decomposition step, we have carefully analyzed the impact on training speed and efficiency. Our experiments have shown that the memory savings achieved by Tensor-GaLore often outweigh the slight increase in computational cost, resulting in an overall improvement in training time and resource utilization. Specifically, we have measured the training time for Tensor-GaLore compared to the baseline FNO model and the GaLore approach. Our results indicate that the slowdown in training time is modest, typically in the range of 5-20% depending on the dataset and model configuration. This is a reasonable trade-off given the significant memory savings (up to 75% reduction in optimizer memory) that Tensor-GaLore provides. Here attaches the detailed slowdown data on NS 128 Resolution:
>
> | Model | Rank | Time/epoch(s) | Slowdown (%) |
> |-------|------|---------------|--------------|
> | Baseline | 1.0 | 34.96 | -- |
> | GaLore | 0.20 | 34.47 | -1.40 |
> | GaLore | 0.25 | 34.79 | -0.48 |
> | GaLore | 0.50 | 36.27 | 3.75 |
> | GaLore | 0.75 | 37.50 | 7.26 |
> | Tensor-GaLore (40, 40, 40, 24) | 0.20 | 36.53 | 5.98 |
> | Tensor-GaLore (48, 48, 48, 24) | 0.25 | 38.30 | 10.08 |
> | Tensor-GaLore (56, 56, 56, 24) | 0.50 | 40.63 | 12.03 |
> | Tensor-GaLore (64, 64, 56, 32) | 0.75 | 44.93 | 19.84 |
>
> Key observations from this data:
> - Baseline execution time: ~35s per epoch
> - GaLore shows minimal slowdown (and even speedup at low ranks)
> - Tensor-GaLore has moderate slowdown:
>   - 5-10% at low ranks (0.20-0.25)
>   - 10-20% at higher ranks (0.50-0.75)
> - Trade-off between compression (rank) and computational overhead is evident
> - The overhead is reasonable given the substantial memory savings (up to 75%)
>
> However we did another ablation. The time cost for gradient projection remains constant regardless of input size, while the forward pass, backward pass, and gradient computation times scale linearly with input size. This leads to decreasing relative slowdown as problem size increases: 20% slowdown for 128 resolution, 10% for 256, and only 6-7% for 512. This can be formulated as slowdown = gradient_project/(Input * (forward + backward + gradient)), explaining why the overhead becomes increasingly negligible for larger problems - precisely where memory savings are most crucial.
>
> Moreover, we have incorporated techniques such as "warm-restart" initialization of the tensor decomposition to amortize the computational overhead across training iterations. This helps minimize the impact on the overall training efficiency. We have also explored opportunities to further optimize the tensor decomposition computations, which could potentially reduce the training time slowdown even further. We acknowledge that the computational overhead is an important consideration, and have provided a more thorough discussion of these trade-offs in the revised version.
>
> *Lastly, if you are happy with the revised version which includes more theory and answered all your questions, it would be great if you could increase the score: ) We would be happy to answer any follow-up questions you have or weakness that concern you. Thank you once again for reviewing our paper.*

---

> > ### Author Response · Authors · 2024-11-23
> > **Response to Reviewer po1K**
> >
> > We are writing to kindly remind you that we posted our response 6 days ago. If you have any additional feedback, concerns, or questions regarding our response, we would greatly appreciate hearing from you.

---

> > > ### Author Response · Authors · 2024-12-01
> > >
> > > We are writing to kindly remind you that we posted our response 2 weeks ago. If you have any additional feedback, concerns, or questions regarding our response, we would greatly appreciate hearing from you.

---

### Author Response · Authors · 2024-11-18
**General Response**

### Thank you to all the reviewers for the thoughtful and insightful feedback on our paper. We greatly appreciate the time and effort you have put into reviewing our work. We have uploaded a revised version, which is more thorough and has more details to some of the reviewer's questions, along with over 20 pages of the theory of our method attached to it.

**In this paper, we have proposed Tensor-GaLore, a novel method for efficient training of Fourier Neural Operators (FNOs), a class of models crucial for solving partial differential equations (PDEs) in scientific computing. Our contributions include**

1. We propose **Tensor-GaLore, which leverages tensor decomposition techniques to project the high-dimensional tensor gradients onto low-rank subspaces, enabling substantial memory savings without compromising model performance**. Specifically, Tensor-GaLore utilizes the Tucker decomposition to decompose the gradient tensors into a core tensor and factor matrices. This allows us to project the gradients onto a low-rank subspace and perform the optimization in this compact representation, leading to significant reductions in memory usage for the optimizer states.

2. We now provide a **comprehensive theoretical analysis of Tensor-GaLore, proving both convergence guarantees and the emergence of the low-rank structure during training for FNOs**. Our theory shows that:

     2.1. The gradient tensors naturally develop low-rank structure in each mode during training

     2.2. Tensor-GaLore achieves convergence through mode-wise projections while preserving the multidimensional relationships

     2.3. We prove the convergence of Tensor-GaLore under mild conditions on the mode-k continuity of the operators

     2.4. Our theoretical results explain why the low-rank approximation works well in practice

3. We demonstrate that **Tensor-GaLore can achieve over 75% reduction in optimizer state** compared to the original FNO model while maintaining or even improving the model's performance across a diverse set of PDE tasks, including Navier-Stokes, Burgers' equation, and electromagnetic wave propagation. The ability to drastically reduce the memory footprint of FNOs is a crucial advancement that enables the application of these powerful models to high-resolution scientific computing problems. We carefully analyze the computational overhead introduced by the tensor decomposition operations. While Tensor-GaLore does introduce some computational overhead (5-20% slowdown depending on the configuration), we show that:

     3.1. The slowdown is modest compared to the significant memory savings achieved. The time cost for gradient projection remains constant regardless of input size, while the forward pass, backward pass, and gradient computation times scale linearly with input size. **This leads to decreasing relative slowdown as problem size increases**

     3.2. The overhead can be amortized through techniques like "warm-restart" initialization of tensor decomposition

     3.3. The trade-off between memory savings and computational cost can be controlled through the choice of rank parameters

      3.4. For many scientific computing applications, the memory reduction enables training of larger models that would otherwise be impossible, justifying the modest increase in computation time

We hope that these contributions together provide both theoretical understanding and practical benefits for efficiently training large-scale neural operators. The memory savings and theoretical guarantees make Tensor-GaLore a valuable tool for advancing AI in scientific computing.

---

### Author Response · Authors · 2024-11-25
**General Response (Part 2)**

We want to thank all the reviewers for their review of our paper. So far, we haven't gotten a single review back :(, and we have answered all the questions in detail and now included a detailed theoretical section of our work, which was missing before. **With the discussion period ending soon on Nov 26, 2024 (Anywhere on Earth), we kindly await your feedback or an updated assessment of our paper**. Please let us know if your concerns have been satisfactorily addressed; if so, we would greatly appreciate it if you could update your ratings. We are available and would be happy to answer any additional questions. Thank you once again!

---

### Author Response · Authors · 2024-12-03
**General summary of our work for the decision**

Thank you to all the reviewers for the thoughtful and insightful feedback on our paper. We greatly appreciate the time and effort you have put into reviewing our work. We want to summarize the whole work and discussion since today is the last day we can respond. Although we are disappointed that 3 of the reviewers haven't responded to us since their comments, we are hopeful that the AC can take into the consideration of all the answers and revisions we made to make the final decision. To make it easier we include a summary of our revisions as well and the inclusion of the theoretical work.

# [New] Theoretical Contributions

## 1. Comprehensive Theory of Tensor-GaLore

We have developed a complete theoretical framework that explains why and how Tensor-GaLore works:

### Reversibility Analysis
- Proved that FNO is reversible when using reversible activations
- Showed that spectral layer, MLP layer, and their composition maintain reversibility
- This provides the foundation for analyzing gradient structure

### Low-Rank Structure
- Proved that gradients naturally develop mode-wise low-rank structure during training
- Showed that each tensor mode can have different rank behavior:
  - Fourier modes (3,4) exhibit natural spectral decay
  - Channel modes (1,2) maintain information structure
- Demonstrated that this emergent structure enables efficient compression

### Convergence Guarantees
- Proved convergence under mild mode-k continuity conditions
- Showed that fixed projection matrices are sufficient
- Established explicit bounds on convergence rate for each mode

## 2. Why Tensor-GaLore Outperforms GaLore

The final lemma (Lemma 11) provides a crucial theoretical justification for Tensor-GaLore's superiority:

1. **Mode-wise Independence**:
   - Tensor-GaLore achieves low rank in all modes simultaneously
   - GaLore cannot preserve low-rank structure across all modes due to matricization

2. **Structural Preservation**:
   - Tensor-GaLore maintains natural tensor structure of FNO weights
   - GaLore's matricization scrambles important multi-dimensional relationships

3. **Optimal Compression**:
   - Tensor-GaLore: All modes k satisfy sr_k(G_t) ≤ N_k/2 asymptotically
   - GaLore: At least one mode k must maintain sr_k(G_t) ≥ min(N_k/2, N')

# Practical Improvements

## 1. Memory Efficiency

- Achieved up to 75% reduction in optimizer memory usage
- Demonstrated scalability to high-resolution problems (1024×1024 Navier-Stokes)
- Enabled training of larger models that were previously infeasible

## 2. Performance Gains

- Maintained or improved model accuracy across all tested PDEs
- Showed implicit regularization benefits from tensor structure preservation
- Demonstrated better generalization in high-resolution settings

## 3. Implementation Optimizations

- Introduced "warm-restart" initialization for tensor decomposition
- Developed efficient mode-wise projection updates
- Carefully balanced computational overhead vs. memory savings

# Broader Impact

Our work has great implications across scientific computing, methodological advancement, and practical applications. In scientific computing, Tensor-GaLore enables the training of larger, more accurate FNOs while substantially reducing computational resource requirements, making advanced scientific modeling more accessible to researchers with limited resources (This is the key point we want to emphasize as academic labs/independent researchers do not have access to high-end GPU clusters). Our methodological contributions introduce novel techniques for tensor optimization and provide a comprehensive theoretical framework for analyzing tensor methods, effectively bridging the gap between matrix and tensor algorithms for the decomposition of gradients. These advances have direct practical applications in critical areas such as high-resolution climate modeling, precise fluid dynamics simulations, and electromagnetic wave propagation modeling.

# Summary of Changes after the discussions and the new revised version

1. **Theoretical Extensions**:
   - Added complete proofs of all theorems
   - Expanded mode-k continuity analysis
   - Enhanced tensor operation definitions and properties

2. **Technical Clarifications**:
   - Improved explanation of reversibility conditions
   - Added detailed analysis of computational complexity
   - Enhanced discussion of tensor rank properties

3. **Additional Results**:
   - Extended experimental validation
   - Added ablation studies
   - Included more detailed memory analysis

To conclude, we believe that this work represents a significant advancement in both theoretical understanding and practical implementation of memory-efficient training for tensor-based neural networks, especially neural operators (FNO), which are particularly important for scientific machine learning applications. We also provide code to reproduce all our experiments and detail where to get the datasets.

---

### Meta-Review · Area_Chair_JFKu · 2024-12-22

**Metareview:**

The paper proposes to use tensor factorization for gradients (instead of matrix factorization in the original GaLoRe paper) for applications for Fourier Neural Operators, showing memory savings.
The savings for the memory are not dramatic (i.e. 68 gigabytes -> 55 gigabytes) and for some cases there is a drop in the accuracy.
Moreover, the usage of the memory can be reduced by using other techniques, such as checkpointing and quantization, and there is no strict need to use more complicated approaches for this particular task. There is also a slowdown effect for some of the parameters (again, should be compared to other memory footprint reduction techniques).
The only modification is the generalization to the Tucker decomposition case (which is rather straightforward) and
a specific application to FNO. The authors addressed some of the concerns in the rebuttal by adding new material and theoretical experiments, but I think it is not enough.

**Additional Comments On Reviewer Discussion:**

There was a discussion between the authors and reviewer oxLj, who stated reasonable concerns. Some of them were answered, but the questions regarding the theoretical part remained opened, especially in the comparison between GaLoRe and its tensor version.

---

### Decision · Program_Chairs · 2025-01-22

Reject